SPECIAL ISSUE
CILIA AND FLAGELLA: FROM BASIC BIOLOGY TO DISEASE

# Urinary renal epithelial cells can be used for *NPHP1* phenotyping and a personalized therapeutic strategy

Praveen Dhondurao Sudhindar[1], Eric Olinger[1,2], Zachary T. Sentell[3], Holly Mabillard[1], Barbora Dicka[1], Katrina Wood[4], Dominic Rutland[1], Catherine Collins[2], Marco Trevisan-Herraz[1], John A. Sayer[3,5,6,*] and Juliana E. Arcila-Galvis[3]

**ABSTRACT**

Nephronophthisis (NPHP) is a recessive tubulointerstitial nephropathy and a leading genetic cause of kidney failure in children and young adults. The most common genetic cause is a homozygous deletion of *NPHP1*, which encodes nephrocystin-1, a protein essential for primary cilium structure and cell junctions. Using personalized medicine and deep phenotyping, we investigated a family with three siblings carrying a homozygous *NPHP1* deletion. We compared kidney biopsy tissue and human urine-derived renal epithelial cells (hURECs) from these individuals. Bulk RNA-seq on patient hURECs revealed altered expression in EGFR signalling, extracellular components and adherens junctions, which is consistent with the known roles for nephrocystin-1. Treatment with alprostadil, a proposed NPHP therapy, increased ciliation but worsened ciliary elongation. By contrast, the EGFR kinase inhibitor AG556 rescued of ciliary length and morphology. Transcriptional profiling post-treatment showed AG556 reversed the disease signature more effectively that alprostadil. These findings suggest that EGFR inhibition might offer a more promising therapeutic strategy for *NPHP1*-associated renal ciliopathy, warranting further testing in *in vivo* models before clinical application.

**KEY WORDS: Kidney failure, Nephronophthisis, NPHP1, Whole gene deletion, Primary cilia, RNA-seq, Urine-derived renal epithelial cells, Extracellular matrix, Adherens junctions, Precision medicine**

**INTRODUCTION**

Nephronophthisis (NPHP) is an autosomal recessive tubulointerstitial nephropathy classified as a renal primary ciliopathy disorder and is recognized as the most common genetic cause of kidney failure in children and young adults (Simms et al., 2009). It is characterized by progressive tubulointerstitial fibrosis, corticomedullary cyst formation, tubular atrophy and kidney failure. Clinical manifestations often include polyuria, polydipsia, anaemia and growth retardation, with no significant proteinuria or hypertension in the early stages. The clinical presentation can therefore be insidious, and diagnosis can often be delayed until after kidney failure has been reached. NPHP remains an underdiagnosed condition in kidney failure populations (Snoek et al., 2018; Stokman et al., 2018). Among the known genetic causes, *NPHP1* is the most frequently implicated gene; *NPHP1* encodes nephrocystin-1, a protein essential for maintaining the structure and function of the primary cilium and that has roles beyond the cilium in maintaining cellular junctions and tubular basement membranes (Wolf and Hildebrandt, 2011). Disruptions in nephrocystin-1 are proposed to lead to defects in ciliary signalling, cell polarity and tubular integrity, ultimately contributing to kidney dysfunction. Given the lack of curative treatments, ongoing research focuses on understanding the molecular mechanisms of NPHP and exploring potential therapeutic strategies through personalized medicine approaches and deep phenotyping.

Genetic investigation of kidney diseases via next-generation sequencing approaches has become increasingly more common and has the potential for large diagnostic yields (Groopman et al., 2019); thus, it is hoped that cases of NPHP should be more readily identified and any therapeutic window before kidney failure ensues widened. This would allow a longer period of intervention for any novel therapeutics.

In cases of suspected NPHP, a previous study has shown that whole-exome sequencing (WES) is able to detect a causative variant in almost two-thirds of cases (Braun and Hildebrandt, 2017). However, a significant number of cases remain genetically unsolved, and such cases might represent novel genetic causes, pathogenic deep intronic variants that are filtered out (Larrue et al., 2020), or incomplete detection of structural variants (SVs) (Pagnamenta et al., 2024) and copy number variants (CNVs) (Mansilla et al., 2021). CNV are increasingly recognized to be pathogenic (Louw et al., 2023), and software tools to decipher CNVs across WES and whole-genome sequencing (WGS) datasets are being devised but are not completely accurate (Masood et al., 2024). This is especially relevant for the analysis of *NPHP1* variants, where whole-gene deletions account for over two-thirds of all NPHP cases (Hildebrandt et al., 1997). In fact, the identification of *NPHP1* whole-gene deletions are clinically important and are being recognized as a cause of kidney failure of unknown cause in children and adults (Snoek et al., 2018; Stokman et al., 2018). The Genomics England 100,000 Genomes Project has proved to be a fertile ground for undertaking genotype–phenotype studies for *NPHP1* whole-gene deletions (Leggatt et al., 2023) and providing a resource in which to develop and refine molecular diagnostics (Olinger et al., 2024).

[1]Translational and Clinical Research Institute, Faculty of Medical Sciences, Newcastle University, Central Parkway, Newcastle upon Tyne, NE1 3BZ, UK. [2]Center for Human Genetics, Cliniques Universitaires Saint-Luc, Brussels, Belgium. [3]Biosciences Institute, Faculty of Medical Sciences, Newcastle University, Central Parkway, Newcastle upon Tyne, NE1 3BZ, UK. [4]Royal Victoria Infirmary, Newcastle upon Tyne, NE2 4LP, UK. [5]Renal Services, Newcastle Upon Tyne Hospitals NHS Foundation Trust, Newcastle upon Tyne, NE7 7DN, UK. [6]NIHR Newcastle Biomedical Research Centre, Newcastle upon Tyne, NE4 5PL, UK.

*Author for correspondence ( john.sayer@newcastle.ac.uk)

E.O., 0000-0003-1178-7980; Z.T.S., 0000-0001-6005-7111; H.M., 0000-0001-6725-7570; D.R., 0009-0001-2300-8692; J.A.S., 0000-0003-1881-3782; J.E.A.-G., 0000-0002-7293-2187

When pathogenic variants in *NPHP1* were first described as a molecular genetic cause of NPHP, it was noted that nephrocystin-1 contained an Src homology 3 (SH3) domain (Hildebrandt et al., 1997). This domain was shown to interact with p130Cas, an essential component of focal adhesions, which led to speculation that nephrocystin-1 was involved in the regulation of cell matrix adhesions (Benzing et al., 2001). Indeed, nephrocystin-1 localized with E-cadherin at the adherens junctions of cell–cell contacts in Madin-Darby canine kidney (MDCK) cells (Donaldson et al., 2000). Further research documented nephrocystin-1 in a protein complex containing p130Cas (also known as BCAR1), tensin and proline-rich tyrosine kinase (PYK2; also known as PTK2B). Research has suggested that nephrocystin-1 facilitates recruitment of PYK2 to cell matrix adhesions in response to extracellular signals. Disruption of the focal adhesion complexes might result in defects leading to NPHP, given the histological descriptions of tubular membrane disruption, tubular ectasia and interstitial inflammation. The identification of *NPHP2*, also known as *INVS* (encoding inversin), as the second gene to cause NPHP allowed the interaction of nephrocystin-1 and inversin to be identified, and it was found these two proteins colocalize in the primary cilia of renal tubular cells (Otto et al., 2003). This also led to the association of cilia (and inversin) in the maintenance of planar cell polarity within the noncanonical Wnt signalling pathway (Simons et al., 2005). These findings established the primary cilia as being involved with the pathogenesis of NPHP and correlated with data concerning the role of primary cilia in the pathogenesis of human polycystic kidney disease (Nauli et al., 2003). Indeed, the localization of nephrocystin-1 to the ciliary base and its interaction with other ciliary proteins to regulate the formation and maintenance of primary cilia is fundamental to our current understanding of the pathogenesis.

Since then, there has been an intense focus on NPHP and related disorders as primary renal ciliopathies. More recently, using cellular and murine models, potential treatments for *NPHP1*-related kidney disease have been proposed, including alprostadil (ALP), which has been shown to rescue ciliogenesis and increase ciliary length (Li et al., 2021; Garcia et al., 2022). Epidermal growth factor receptor (EGFR), a membrane tyrosine kinase receptor expressed in the kidney, is activated after renal damage, leading to cell proliferation and kidney fibrosis, and preclinical studies have provided evidence for its potential as a therapeutic target in polycystic kidney disease (Gattone et al., 1995; Veizis and Cotton, 2005; Ryan et al., 2010).

Here, we report a non-consanguineous family and three affected individuals presenting with clinical features of NPHP. The identification, from evaluation of WGS data, of a homozygous *NPHP1* whole-gene deletion in all three affected siblings provided a precise molecular genetic diagnosis for the family. Using native kidney biopsy tissue from the proband, we were able to determine the primary renal ciliary phenotype in renal tubules using immunofluorescence studies. The primary ciliary phenotype was then shown to be replicated in human urine-derived renal epithelial cells (hURECs) from an affected younger sibling. The transcriptomic disease signature was explored using hURECs from individuals with NPHP, and this allowed the identification of disease pathways, including cell cycle, cell adhesion and EFGR signalling. We characterized the response of ALP treatment on the primary renal cilia of hURECs from individuals with NPHP, which produced an exaggerated ciliary phenotype and an incomplete rescue of the transcriptomic disease signature. In contrast, treatment of the same hURECs with EGFR inhibitor resulted in the rescue of primary cilia phenotypes and a reversal of disease signatures at the transcriptomic level. Together, these data support the use of EGFR inhibition as a potential treatment for renal ciliopathies, which should now be evaluated in *in vivo* models of NPHP before progressing to human studies.

## RESULTS
### Family studies
The index case was a 12-year-old female who was admitted to the hospital with abdominal pain and fever. Routine biochemistry investigations revealed a raised serum creatinine (180 µmol/l, eGFR 30 ml/min/1.73 m$^2$, revised Schwartz equation; Schwartz et al., 2009). Subsequent investigations revealed normal-sized but echogenic kidneys on renal ultrasound scanning, a normal dimercaptosuccinic acid (DMSA) radioisotope scan and a bland urine (negative for blood and protein). The individual had no known family history of kidney disease, and the only other past medical history was a diagnosis of autism. There was no known consanguinity (Fig. 1A). The kidney function did not improve with supportive care, and a renal biopsy was performed on the proband (II:2), which demonstrated severe chronic tubule interstitial damage and thickening and thinning of the tubular basement membranes, consistent with NPHP (Fig. 1B,C). Following informed consent, WGS via the Genomics England 100,000 genomes project was performed on the proband (II:2) DNA together with DNA from both parents. Within a few weeks of this presenting episode in the proband, her older brother presented (at the age of 17 years) to the same hospital with severe lethargy and breathlessness. There had been no other previous hospital admissions, and the only past medical history of note was a diagnosis of autism. Investigations revealed kidney failure (serum creatinine 1158 µmol/l, eGFR 5 ml/min/1.73 m$^2$ Hb 106 g/l). A computed tomography (CT) scan of the abdomen revealed a normal appearance of the kidneys with no evidence of cystic kidney disease. A renal biopsy was not performed given the known histological findings of NPHP in his sister and the established kidney failure. He was immediately commenced on haemodialysis for renal replacement therapy.

Given the presentation of the two older siblings, the youngest sibling (II:3), aged 8 was then tested for evidence of kidney disease. Serum biochemistry at this point in time (and urinary concentration ability) was preserved, but the need for a precise molecular genetic diagnosis in the family became increasingly important, in terms of determining whether the youngest sibling was likely to be affected and testing potential live-related kidney donors prior to living donor kidney transplantation.

Following a negative genetic report from Genomics England regarding the proband (II:2), we undertook a bespoke reanalysis of the WGS data. Using a candidate gene approach, we failed to identify pathogenic variants in coding regions of known NPHP genes. We then looked for CNV changes across known NPHP genes, including *NPHP1* on chromosome 2. This targeted analysis using WGS data was performed using data from the proband and both parents. Using this approach, we identified a homozygous whole-gene deletion of *NPHP1* in the proband, which segregated from each parent. This genetic finding was consistent with the clinical presentation of progressive kidney failure and the renal histology. Whole-gene deletion of *NPHP1* has been associated with both NPHP and neurodevelopmental disorders, including autism spectrum disorder (ASD) (Sakakibara et al., 2022). Autozygosity mapping did not show any large homozygous regions consistent with remote consanguinity (Fig. 1D). The homozygous *NPHP1* whole-gene deletion is clearly shown by loss of read depth with WGS BAM files visualized using IGV (Fig. 1E), and was confirmed in the proband and the affected sibling using multiplex ligation-dependent probe amplification (MLPA). It also allowed screening

Journal of Cell Science

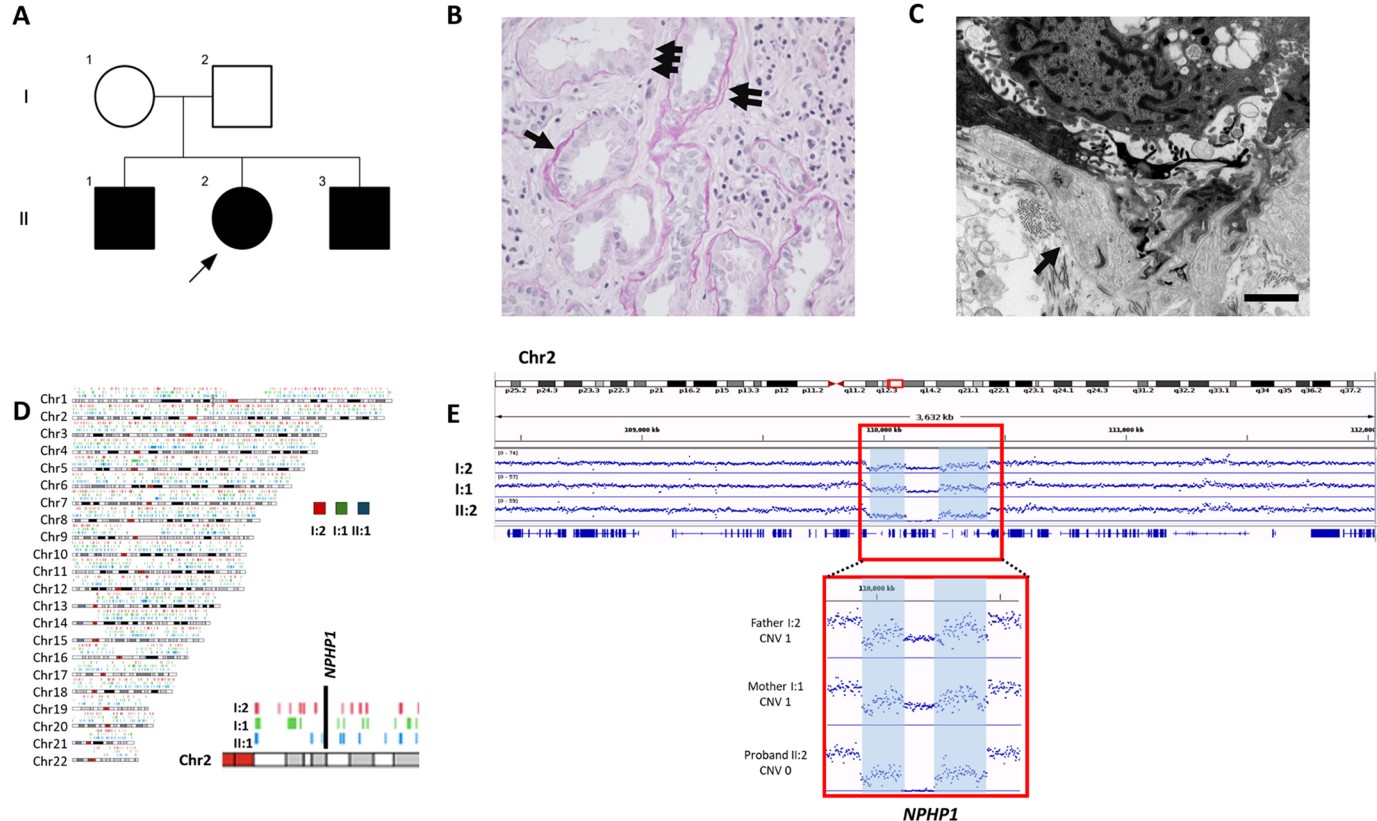

**Fig. 1. Clinical and genetic investigation of a family with nephronophthisis.** (A) Pedigree diagram showing three affected siblings with progressive chronic kidney disease. The arrow shows the proband (II:2). (B) Renal biopsy from II:2. Periodic acid-Schiff-stained section shows variation in thickness of tubular basement membranes with thick areas (single arrow), lamellated areas (double arrow) and areas of absence (triple arrow). Image taken at ×400. (C) Renal biopsy from II:2. Electron microscopy image showing lamellation of tubular basement membranes (arrow). Scale bar: 2 µm. (D) Autozygosity plots for the parent–child trio with detail of the chromosome 2 region containing *NPHP1*, showing no evidence for autozygosity by descent. (E) Average alignment plots for father-mother-child (I:2; I:1; II:2) trio demonstrating *NPHP1* whole-gene deletion in the affected II:2 individual. Average alignment data were generated from BAM files using the count application in IGVtools, generating tiled data files. In the hg38 genome assembly, the deletion spans at least from approximately chr2:109,931,849 to chr2:110,441,473. The regions flanking this deletion appear as low-mappability regions (shaded pale blue). Upstream, low mappability extends toward chr2:109,931,849, and downstream, it extends to around chr2:110,441,473. Inspection of the UCSC Genome Browser (hg38 assembly) shows that these regions correspond to segmental duplications (low-copy repeats) (see Fig. S1). Owing to these segmental duplications, the precise deletion breakpoints in each parent cannot be determined, and it remains unclear whether the maternal and paternal deletion alleles are identical in size or position.

and detection of the *NPHP1* whole-gene deletion in the unaffected younger sibling, prompting paediatric referral and monitoring.

## Characterization of cellular phenotypes in kidney epithelial cells from individuals with *NPHP1* whole-gene deletion

Immunofluorescence staining of cilia using antibodies to ARL13B in a kidney biopsy from the proband (II:2) with a *NPHP1* whole-gene deletion and a sex- and age-matched control (Fig. 2A,B), revealed that primary cilia in the individual with NPHP were significantly longer than those observed in the control biopsy (Fig. 2C). Additionally, ciliary tortuosity was significantly increased in the *NPHP1* tissue (Fig. 2D). There was no change in ciliation rates in renal epithelial cells between *NPHP1* cells and control. The increased ciliary length and tortuosity phenotype in the *NPHP1* individual was seen in collecting duct segments of the nephron identified by aquaporin 2-positive nephron segments in the kidney biopsy tissue for the individual with NPHP (Fig. S2).

Owing to the rapid progression of kidney disease and subsequent kidney transplantation in sibling II:1, hURECs were not available for analysis. However, for the youngest sibling, individual II:3, who had not yet reached kidney failure, we were able to obtain fresh urine samples. From these, we derived primary hURECs and analysed their phenotype

(Fig. S3). Immunofluorescence staining of cilia using antibodies to ARL13B in sibling II:3 hURECs (Fig. 3B,D,F) and hURECs derived from a matched control (Fig. 3A,C,E), following serum starvation to promote ciliogenesis, revealed an increased primary cilia length (Fig. 3G) and tortuosity (Fig. 3H) compared to control hURECs. This phenotype closely resembled that of proband II:2 kidney biopsy which had documented collecting duct segment ciliary phenotypes.

## Transcriptomic signatures of hURECs from *NPHP1*-deletion individual II:3

To understand changes in gene expression associated with the aberrant phenotypes in sibling II:3, we compared the transcriptomes of the hUREC samples derived from urine collected at different time points from sibling II:3 and a healthy matched control (Fig. S3). To ensure that the observed changes in gene expression were not due to heterogeneous cell types present in the two sets of samples, we analysed the expression levels of cell type markers (Balzer et al., 2022). These were found to be comparable between the samples (Fig. S4A). Additionally, principal component analysis (PCA) was performed, which showed that, although there was inter-sample variability, the replicates from subject and control were separated (Fig. S4B).

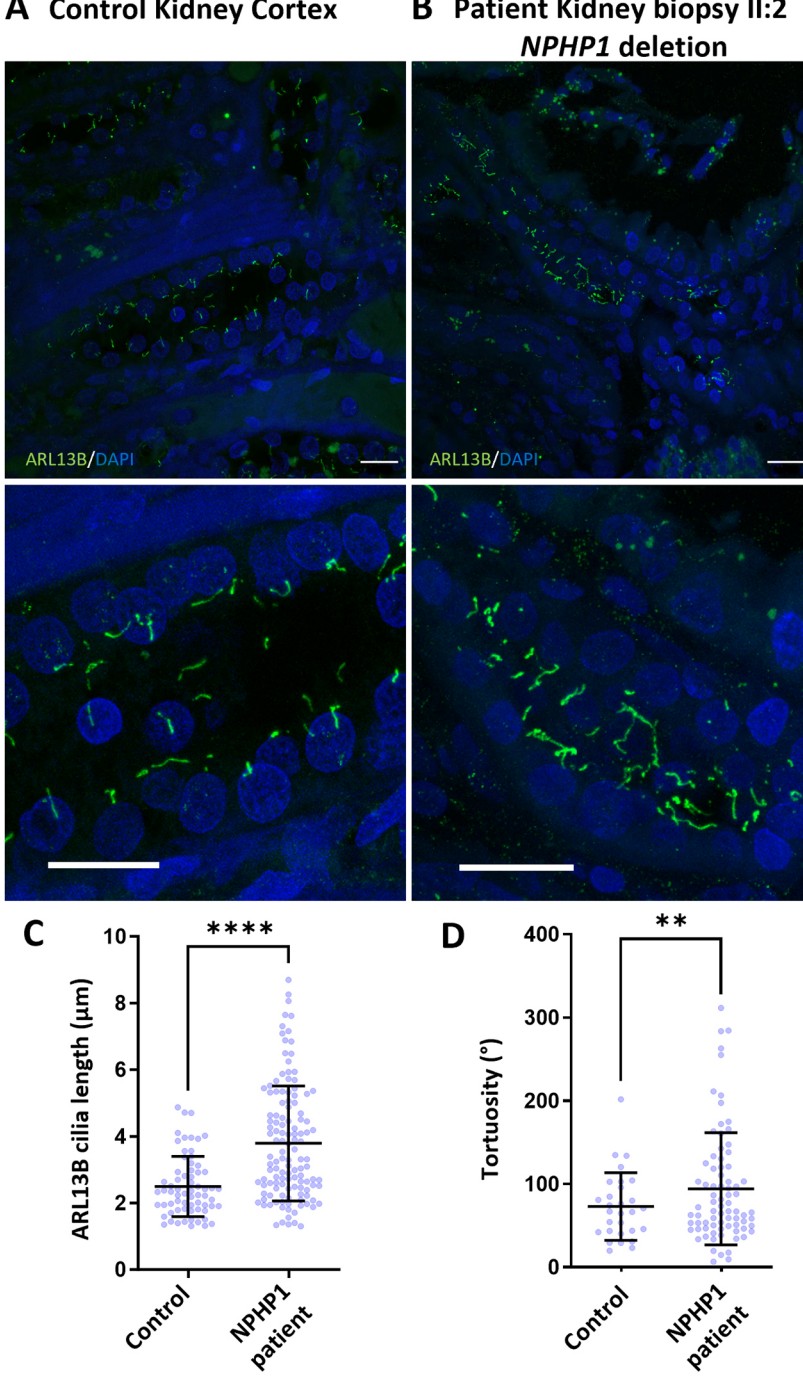

**A** Control Kidney Cortex
**B** Patient Kidney biopsy II:2 *NPHP1* deletion

**C**
**D**

**Fig. 2. Primary cilia analysis in human kidney biopsy tissue from an individual with *NPHP1* whole-gene deletion.** (A,B) Renal biopsy tissue stained with DAPI (blue) and ARL13B (green) in healthy control (A) and individual II:2 (B) with *NPHP1* whole-gene deletion. Scale bars: 20 μm. (C) Quantification of renal tubular cell primary cilia phenotypes. Primary cilia mean length was 2.4 μm in control kidney tissue ($n$=215) and 3.8 μm in patient II:2 kidney tissue ($n$=128). Error bars are mean±s.d. ****$P$<0.0001 (two-tailed unpaired $t$-test). (D) Primary cilia tortuosity was measured and plotted as the sum of angles (°) in control kidney tissue and patient II:2 kidney tissue, **$P$<0.001 (two-tailed unpaired $t$-test).

Differential gene expression analysis identified 956 differentially expressed genes (DEGs) in sibling II:3 hURECs compared to controls, with 448 upregulated and 508 downregulated genes (Fig. 4A; Table S1). The most upregulated gene was *GALNT9*, which initiates O-linked glycosylation, suggesting increased O-GlcNAcylation, a modification known to inhibit autophagy under stress (Chun et al., 2015; Zhu et al., 2024). Other genes among the top upregulated were *SLC6A15* (PEPT1), involved in peptide reabsorption in the proximal tubule (Killer et al., 2021), *VGF*, a stress-responsive protective gene (Kim et al., 2020), and *HAVCR1* (KIM-1), which plays a protective role during acute injury by clearing cellular debris, however; its sustained expression in chronic injury is linked to inflammation and fibrosis (Han et al., 2002; Brooks and

Bonventre, 2015). Among the top downregulated genes were *ITGA11*, involved in type I collagen binding (Bansal et al., 2017), *THBS1*, a matricellular protein regulating adhesion, angiogenesis and immune responses (Isenberg and Roberts, 2020), *PADI2*, a regulator of ROS-induced senescence and immune signalling via NFκB (Liu et al., 2018a; Kim et al., 2022), and *PDGFRB*, which promotes mesenchymal cell proliferation and fibrotic transformation (Buhl et al., 2020) (Fig. 4A). Analysis by quantitative (q)PCR confirmed the transcriptomic findings, with selected genes from key pathways – including integrin signalling, cell proliferation and survival, stress response, basement membrane organization, TGF-β signalling, autophagy, EGF and MAPK – showing expression changes in the same direction as observed in RNA-seq data (Fig. 4B).

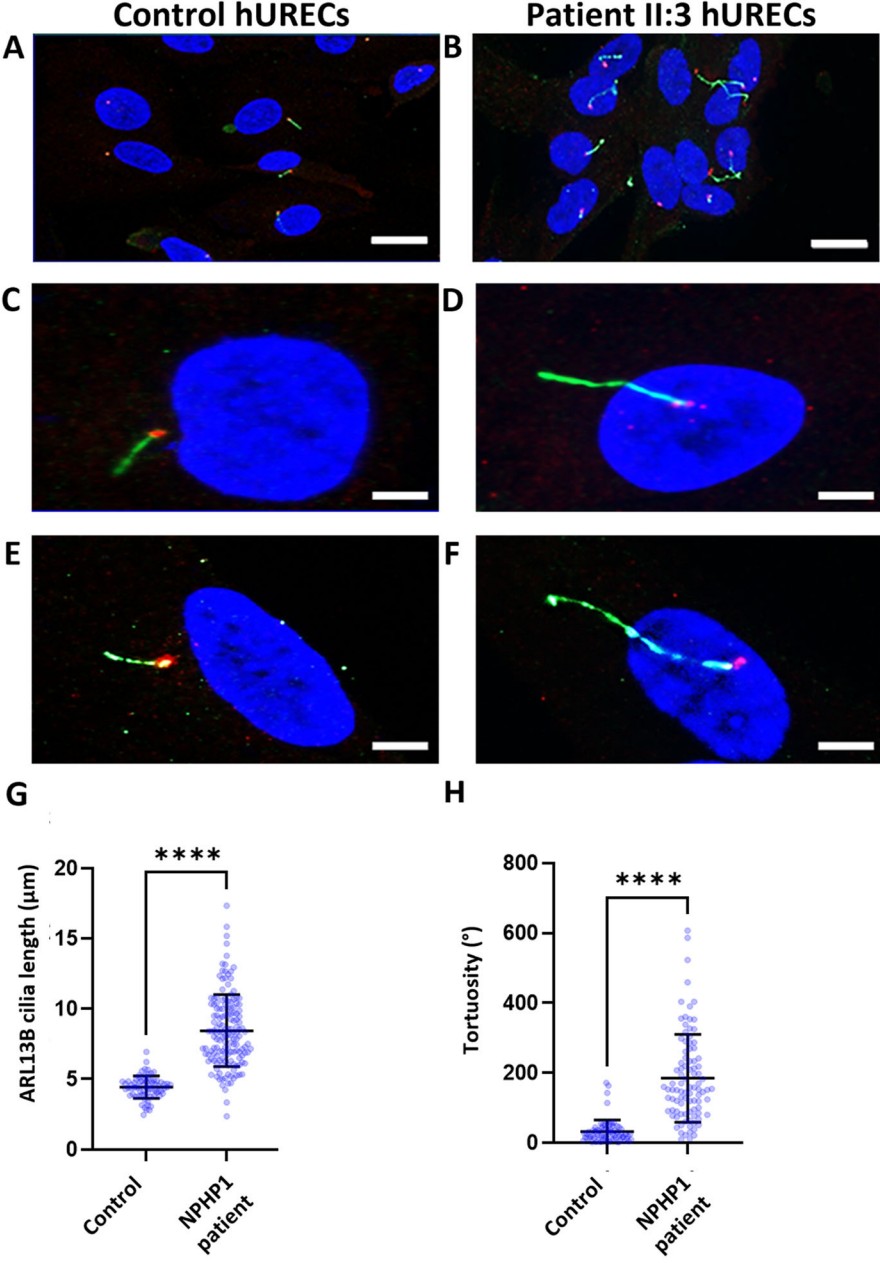

**Fig. 3. Deep phenotyping of renal epithelial cell primary cilia using hURECs from an individual with *NPHP1* whole-gene deletion.** Immunofluorescence microscopy images of (A,C,E) control hURECs and (B,D,F) *NPHP1*-deficient (II:3) hURECs. DAPI (blue); ARL13B (green); γ-tubulin (red). Scale bars: 10 μm (A,B); 5 μm (C–F). (G) Quantification of cilia length. The mean cilia length measured in control hURECs (*n*=71) was 4.4 μm compared to 8.4 μm in II:3 hURECs (*n*=152). Data shown as scatter plot. Error bar indicates mean±s.e.m. ****$P<0.0001$ (two-tailed unpaired *t*-test). (H) Primary cilia tortuosity was measured and plotted as sum of angles (°). The mean tortuosity (°) of the wild-type hURECs (*n*=70) was 32° versus 184° in II:3 hURECs (*n*=94). Error bars shows mean±s.d. ****$P<0.0001$ (two-tailed unpaired *t*-test).

Pathway enrichment analysis of the DEGs, using Gene Ontology term enrichment, revealed significant enrichment in several biological processes including those related to kidney and nephron epithelium development, Notch and phosphoinositide 3-kinase (PI3K)–AKT signalling, regulation of epithelial–mesenchymal transition (EMT), insulin-like growth factor (IGF) transport, and immune cell migration and activation (Fig. 5A). Cellular component analysis showed enrichment in the extracellular matrix, basement membrane and collagen type IV, but only among downregulated genes, with no enrichment detected in the upregulated gene set (Fig. 5B). Pathway enrichment results from KEGG, Reactome and WikiPathways databases identified enrichment of upregulated genes in MAPK, Ras and EGF–EGFR signalling pathways. In contrast, downregulated genes were enriched in VEGFA–VEGFR2 signalling, p53, integrin-mediated interactions, PDGF signalling, collagen degradation and apoptosis pathways. Some pathways, such as receptor tyrosine kinase (RTK) signalling, PI3K–AKT and

AGE–RAGE, included both up- and down-regulated genes (Fig. 5C; Table S3).

Heatmaps displaying the top significantly altered genes in selected enriched pathways highlighted that there are upregulated pathways related to inflammation, actin cytoskeleton dynamics and nephron development (Fig. S5A), and downregulated pathways including basement membrane organization, apoptosis, cell–cell junctions and focal adhesions (Fig. S5B).

Together, these findings indicate that RTK-mediated growth factor signalling is disrupted at multiple levels in *NPHP1*-deficient cells, potentially reflecting a cilia-related trafficking or signalling loss. Specifically, downregulation of VEGFA–VEGFR2, PDGF signalling and IGF transport pathways, all of which are associated with RTK activity, suggests impaired regulation of vascular function, mesenchymal cell activation, fibrosis control and growth factor availability. In contrast, upregulation of EGF–EGFR, MAPK, Ras and PI3K–AKT pathways, also downstream of RTKs, indicates

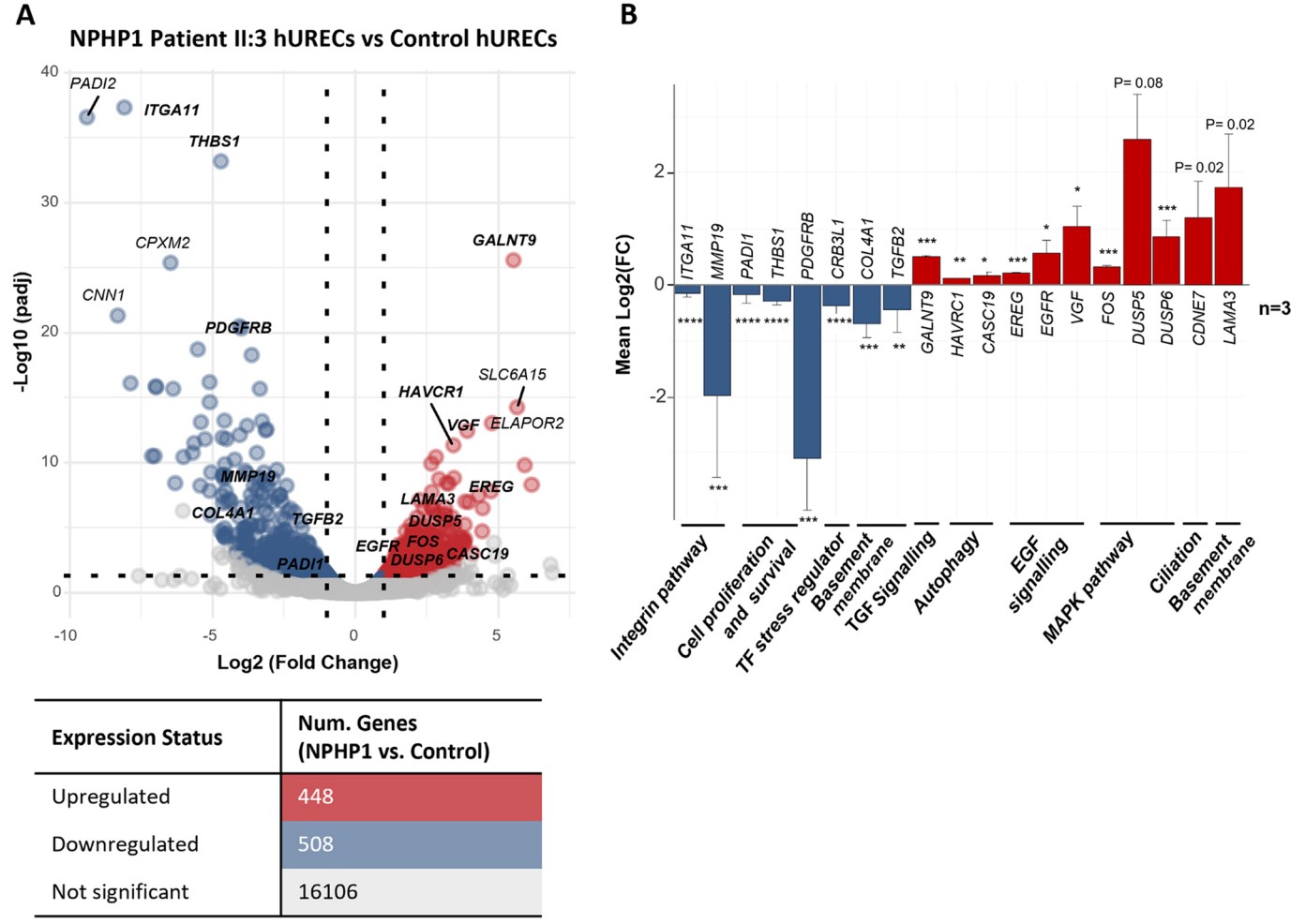

**Fig. 4. Gene expression analysis comparing case and control samples.** (A) Volcano plot showing the differential gene expression distribution, where genes with significant upregulation are shown in red, downregulation in blue and genes with non-significant changes in grey. The table below summarizes the number of upregulated, downregulated, and not significant genes based on differential expression analysis (threshold of |Log$_2$FC|>0.58 and adjusted $P$-value<0.05). Gene labels shown for the top five most significant genes and genes chosen for qPCR validation (in bold). (B) Barplot with the mean fold changes from the qPCR validation of 19 selected dysregulated genes in II:3 hURECs compared to control hURECs ($n$=3). Genes were selected from different relevant pathways. ****$P$<0.0001, ***$P$<0.001, **$P$=0.0019, *$P$=0.0348 (two-tailed unpaired $t$-test). Error bars indicate mean±s.e.m. To enhance visualization, values greater than 2 or less than −2 on the y-axis have been log-transformed using the following formula: log-transformed $y$ value= ±2+log$_2$($y$ value −2).

aberrant activation of proliferation, survival and differentiation programmes.

### Modulation of *NPHP1*-associated cellular phenotypes in primary hURECs using small molecules

To assess whether pharmacological interventions could mitigate the aberrant cellular phenotypes associated with *NPHP1* deficiency, we tested treatment with ALP informed by prior published studies and with an EGFR signalling inhibitor, informed by our present transcriptomic analyses (Fig. S6).

### Alprostadil treatment

ALP, a prostaglandin E1 analogue was previously identified in a high-throughput screen as a compound capable of rescuing *NPHP1*-related defects (Garcia et al., 2022). The screen, performed in mIMCD3 and MDCK cell lines with *Nphp1* expression being knocked down via short hairpin RNA, revealed that PGE1 and PGE2 receptor agonists could restore the reduced percentage of ciliated cells and abnormal migration observed upon *Nphp1* loss. Experiments in immortalized URECS from *NPHP1*-deficient

individuals, as well as in *in vivo* zebrafish and mouse models confirmed rescue with ALP treatment (Garcia et al., 2022).

We therefore treated primary hURECs derived from the *NPHP1*-deficient sibling II:3 with ALP. Consistent with the original findings (Garcia et al., 2022), ALP treatment led to an increase in the percentage of ciliated cells (Fig. 6A–G), but it also increased primary cilia length (Fig. 6H). There was no rescue of tortuosity following ALP treatment (Fig. 6I). Notably, in the primary *NPHP1*-deficient hURECs used in our study, both ciliation and cilia length were already elevated relative to controls at baseline, contrasting with the phenotype reported for immortalized cells in the García et al. study. Consequently, ALP treatment in hURECs from individual II:3 exacerbated these phenotypes rather than correcting them. These data suggest that, although ALP can modulate ciliogenesis, its effects are context dependent and might be counterproductive in *NPHP1*-deficient cells, where ciliogenesis is already abnormally augmented.

### Inhibition of aberrant EGFR signalling by AG556

Transcriptomic pathway enrichment analyses from *NPHP1*-deficient hURECS indicated that RTK-mediated signalling pathways,

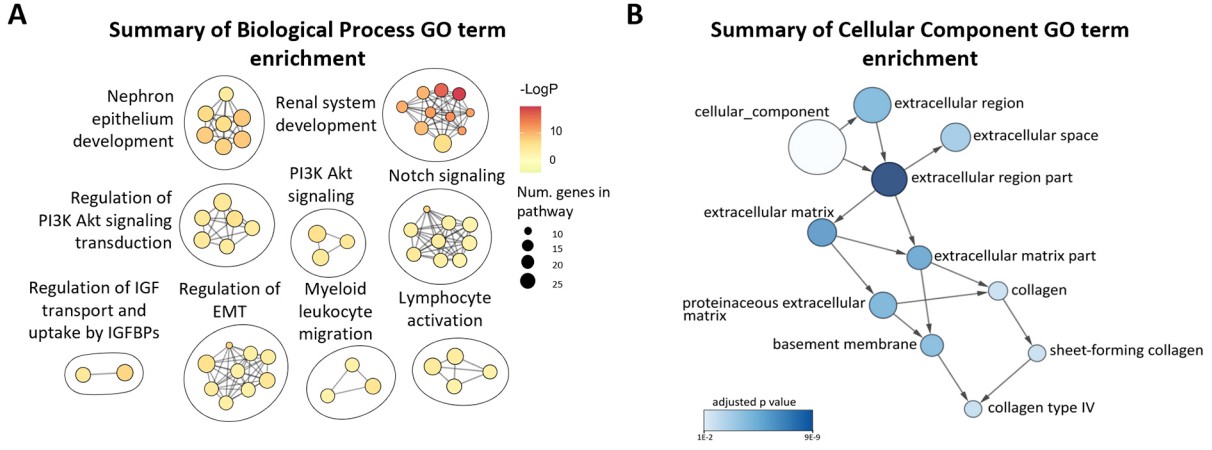

**A** Summary of Biological Process GO term enrichment

**B** Summary of Cellular Component GO term enrichment

**C**

Non-GO term pathway enrichment results for upregulated genes

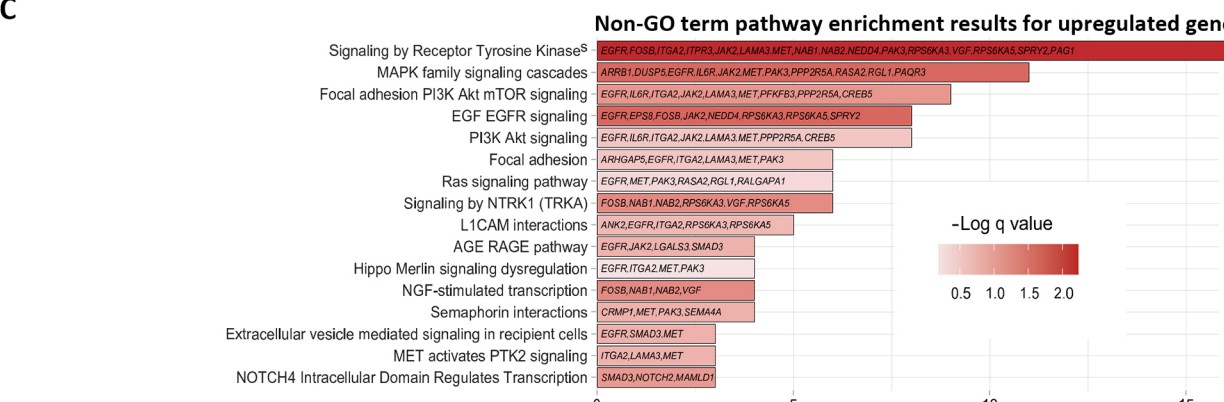

Non-GO term pathway enrichment results for downregulated genes

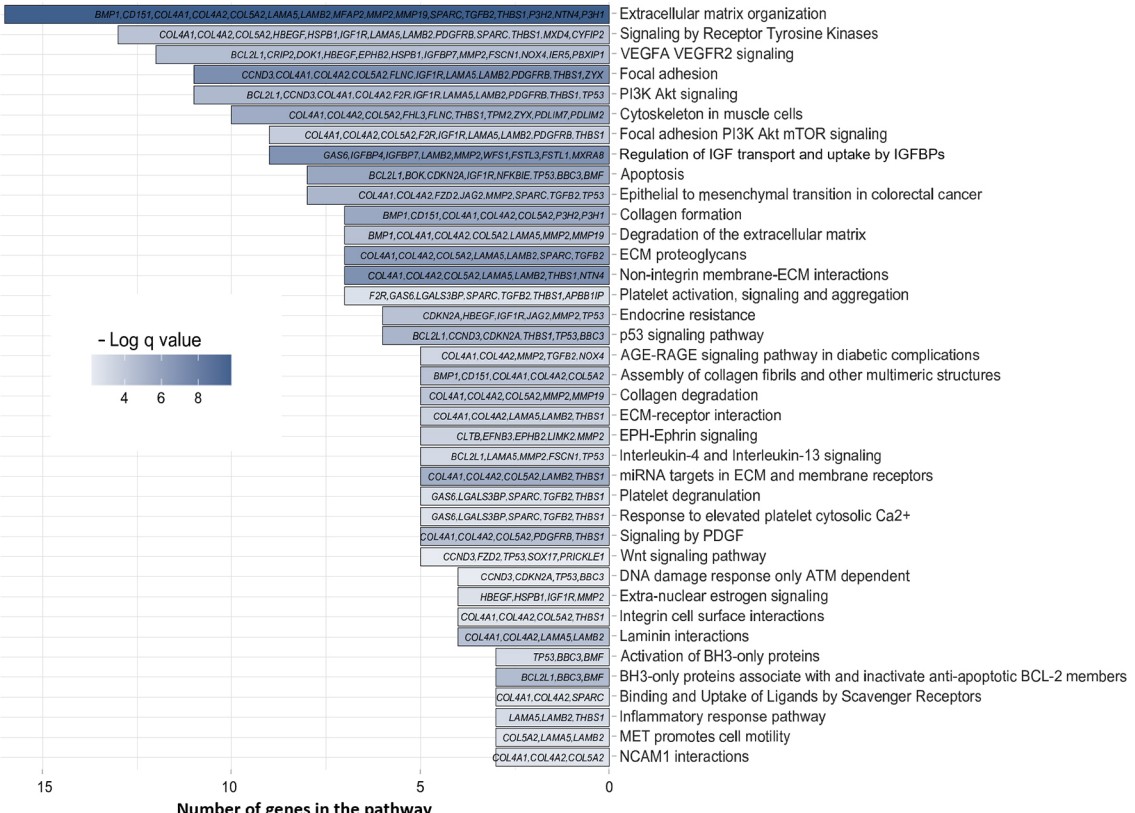

**Fig. 5.** See next page for legend.

**Fig. 5. Pathway enrichment analysis.** (A) Biological Processes GO Term Enrichment Analysis. Each node represents a GO term associated with a biological process, coloured by −log(P-value). (B) Cellular Component GO Term Enrichment Analysis. Each node represents a GO term associated with a cellular component, coloured according to the adjusted P-value. For both A and B, links between nodes represent similarity based on overlap in gene content. Clusters of similar pathways are delineated by ellipses, highlighting redundant or overlapping processes. The size of each node corresponds to the number of genes associated with the GO term. (C) Enrichment results for pathways from Reactome pathways and non-redundant pathways from WikiPathways or KEGG databases. The size of the bars represents the number of genes in the pathway, and the bars are coloured by −log(q-value). Red indicates enrichment of upregulated genes; blue indicates enrichment of downregulated genes.

particularly EGF–EGFR, MAPK, Ras and PI3K–AKT, were upregulated. To test whether this hyperactivation contributed to the observed phenotypes, we treated hURECs with AG556, a selective EGFR tyrosine kinase inhibitor (Levitzki and Gazit, 1995). Treatment of *NPHP1*-deficient hURECs from individual II:3 with AG556 led to a reduction in cilia length and tortuosity, restoring these parameters to levels comparable to those observed in control hURECs. Percentage ciliation was not significantly changed following treatment with AG556 (Fig. 7A–I). These results suggest that inhibition of EGFR

signalling can normalize aberrant ciliary phenotypes in *NPHP1*-deficient cells, where ciliation is already abnormally elevated, supporting a role for dysregulated RTK activity in the disease mechanism. Based on these findings, we propose a model of cilia-associated EGFR signalling pathways and their response to treatments (Fig. 7J). In the wild-type situation, tightly regulated EGFR signalling maintains cell growth, proliferation and survival. There is crosstalk between different ciliary signalling pathways (MAPK, PI3K and Akt) at the cilium centrosome axis to regulate cell cycle, migration, growth and metabolism. *NPHP1*-deficient cells with elongated cilia show an aberrantly activated EGFR pathway. Overexpression of EGFR, and EREG ligand binding to EGFR, alters the MAPK downstream signalling events leading to uncontrolled cell growth and proliferation. Upregulation of DUSP5 and DUSP6, which act as a negative-feedback regulators of ERK signalling, is seen. In cells treated with AG556, a potent inhibitor of EGFR tyrosine kinase, MAPK, PI3K and Akt downstream signalling pathways are suppressed, and normal cilia length is restored.

### Transcriptomic analysis of drug-treated hURECs

To investigate the molecular mechanisms underlying the phenotypic changes observed in small-molecule-treated *NPHP1*-deficient

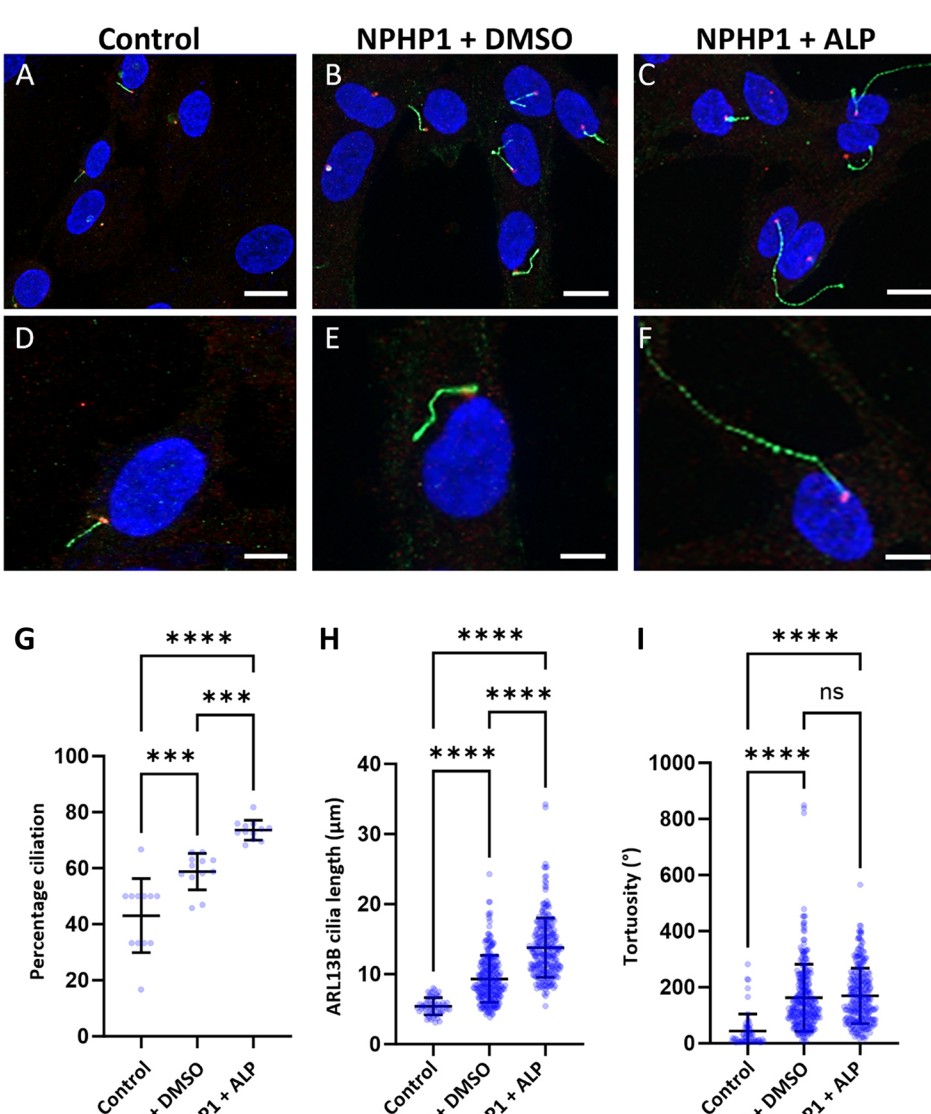

**Fig. 6. Cilia phenotyping of hURECs in response to ALP treatment.** Immunofluorescence microscopy images of control hURECs (A,D) and *NPHP1*-deficient (II:3) hURECs treated with DMSO alone (B,E) and after treatment with alprostadil (ALP) 2 μM in DMSO (C,F). Representative images of low power (A–C) and high-power microscopy (D–F). DAPI, blue (nucleus); ARL13B, green (ciliary membrane); γ-rubulin, red (basal body). Scale bars: 10 μm (A–C); 5 μm (D–F). (G) The mean percentage of cells showing the presence of the primary cilium was measured. 10 cells counted per field, and a total of 12 fields were counted for the control (mean=43%), untreated II:3 hURECs (mean=59%) and II:3 hURECs treated with ALP (mean=74%). (I) Error bar indicates mean±s.d. ****P<0.0001, ***P<0.003 (one-way ANOVA with Tukey's multiple comparisons test). (H) The cilia length was measured in control hURECs (n=58), which was 5.4 μm, compared to the II:3 hURECs (n=283), which was 9.1 μm. Cilia length in II:3 hURECs treated with ALP (n=272) was 13.8 μm. Data shown as scatter plot with means. Error bar indicates mean±s.d. ****P<0.0001 (one-way ANOVA with Tukey's multiple comparisons test). (I) Primary cilia tortuosity (°) was measured and plotted as the sum of angles. The mean tortuosity of control hURECs (n=58) was 44°, compared to 164° in II:3 hURECs (n=283), and 169° in II:3 hURECs treated with ALP (n=272). Error bar indicates mean±s.d. ****P<0.0001; ns, not significant (one-way ANOVA with Tukey's multiple comparisons test).

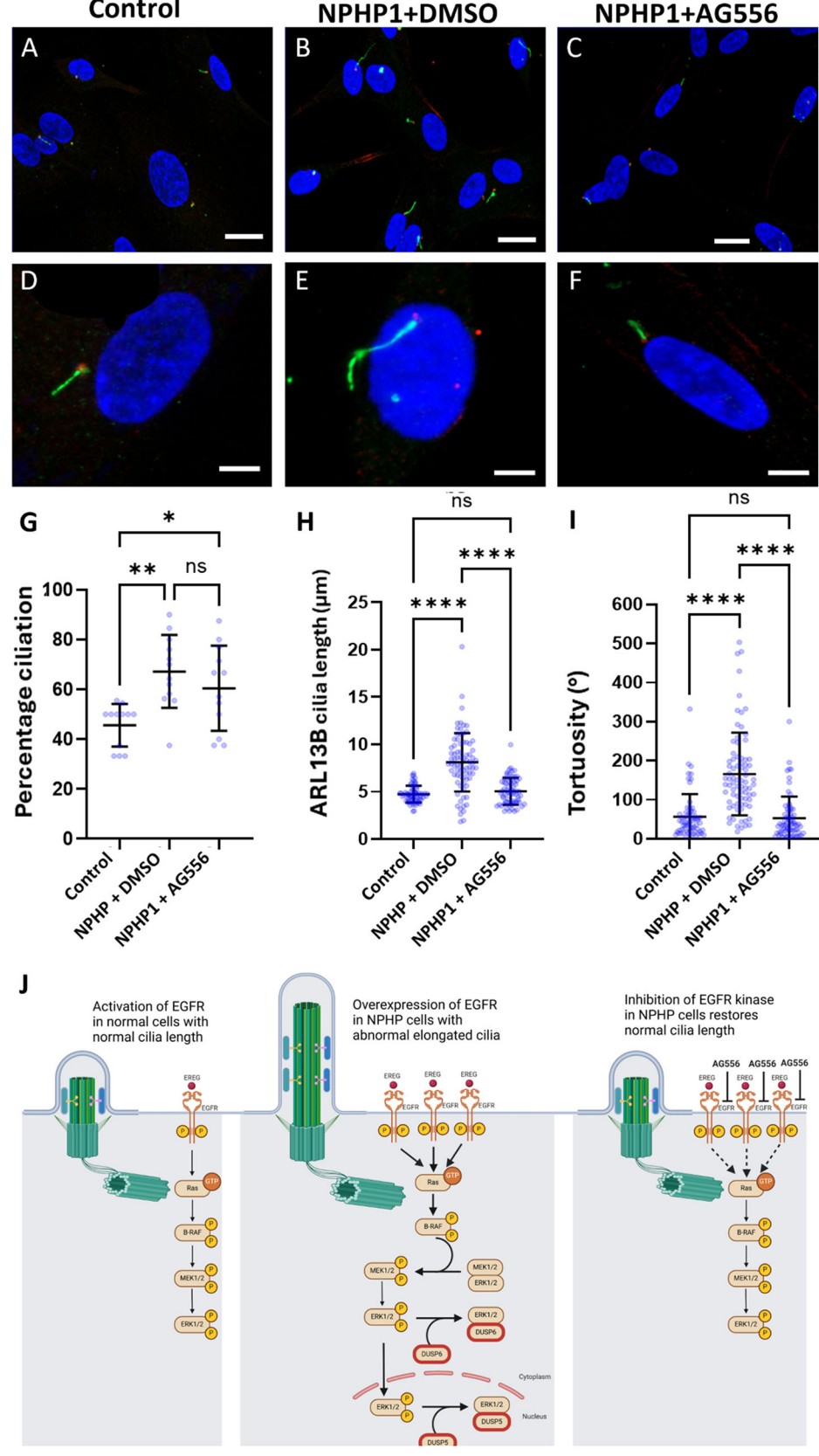

**Fig. 7. Cilia phenotyping of renal epithelial cell primary cilia in response to AG556 treatment and proposed mechanism of action.** Representative low power (A–C) and high power (D–F) immunofluorescence microscopy images of wild type control and *NPHP1*-deficient (II:3) hURECs (DMSO alone) or *NPHP1*-deficient (II:3) hURECs treated with AG556 2 µM in DMSO. DAPI, blue (nucleus); ARL13B, green (ciliary membrane); γ-tubulin, red (basal body). Scale bars: 10 µm (A–C); 5 µm (D–F). (G) Mean percentage of cells with a primary cilium was measured for each individual field. 10 cells counted per field and a total of 12 fields were counted for the wild-type control (mean=46%), II:3 hURECs untreated (mean=67%) and II:3 hURECs treated with AG556 (mean=61%). Error bar indicates mean±s.d. **$P$=0.0016, *$P$=0.0344; ns, not significant (one-way ANOVA with Tukey's multiple comparisons test). (H) The mean cilia length measured for the wild-type control hURECs ($n$=63) was 4.8 µm compared to the II:3 hURECs ($n$=78) was 8.1 µm and treated with AG556 ($n$=67) was 5.1 µm. Data shown as scatter plot with means. Error bar indicates mean±s.d. ****$P$<0.0001; ns, not significant (one-way ANOVA with Tukey's multiple comparisons test). (I) Individual primary cilia tortuosity was measured and plotted as sum of angels (°). The mean tortuosity of the wild-type hURECs ($n$=63) was 57° compared to the II:3 hURECs ($n$=78) was 166° and treated with AG556 ($n$=67) was 53°. Error bar indicates mean±s.d. ****$P$<0.0001; ns, not significant (one-way ANOVA with Tukey's multiple comparisons test). (J) Schematic representation of cilia-related EGFR signalling pathway. Healthy cells (left) show tightly regulated EGFR signalling, maintaining cell growth, proliferation and survival. There is crosstalk between different ciliary signalling pathways (MAPK, PI3K and Akt) at the cilium centrosome axis to regulate cell cycle, migration, growth and metabolism. *NPHP1*-deficient cells (middle) with elongated cilia show an aberrantly activated EGFR pathway. Overexpression of EGFR and EREG ligand binding to EGFR in the alters the MAPK downstream signalling events leading to uncontrolled cell growth, and proliferation. Upregulation of DUSP5 and DUSP6, which act as a negative-feedback regulator of ERK signalling, is seen. In cells treated with AG556, a potent inhibitor of EGFR tyrosine kinase (right), MAPK, PI3K, Akt downstream signalling pathways are suppressed and normal cilia length is restored.

hURECs, we performed transcriptomic analysis on cells treated with either ALP or AG556 and DMSO-treated controls (Fig. S7A–E). The DMSO treatment did not significantly affect differential gene expression, with only ten genes showing significant changes compared to expression in untreated hURECs (Fig. S7C, Table S4).

In the comparison between untreated and ALP-treated *NPHP1*-deficient hURECs, we identified 1637 upregulated genes and 1837

downregulated genes (Fig. S7D, Table S5). In contrast, AG556 treatment resulted in 1638 upregulated genes and 2124 downregulated genes (Fig. S7B, Table S6).

Transcriptomic analysis of DEGs with low variability between time-point replicates per individual (lfcSE<1) revealed several key dysregulated pathways in response to both ALP and AG556 treatments (Fig. S8A,B, Tables S7 and S8). Both drug treatments led to significant changes in gene expression related to Rho GTPase signalling, response to stress, response to external stimuli, cilium biogenesis, cell cycle, mitotic processes and carbohydrate metabolism. ALP-treated cells predominantly showed downregulation of genes in these pathways, whereas AG556-treated cells displayed a mixed response, with a predominance of downregulated genes and a smaller proportion of upregulated genes. Notably, only in the ALP-treated cells did we observe substantial enrichment in protein modification pathways, including asparagine N-linked glycosylation and neddylation, as well as pathways related to lipid metabolism and autophagy (Fig. S8A). Autophagy is known to promote ciliation and cilia elongation (Pampliega et al., 2013; Tang et al., 2013), consistent with the different ciliary phenotypes observed between the two treatments.

We also investigated whether treatment with either ALP or AG556 could modulate the expression of genes dysregulated in *NPHP1*-deficient hURECs from sibling II:3 towards levels observed in control hURECs. To do this, we compared the set of DEGs between untreated *NPHP1*-deficient hURECs and controls to those altered in *NPHP1*-deficient hURECs in response to ALP or AG556 treatment (Fig. 8A; Table S9).

When comparing the disease signature with drug-induced changes, we found that most overlapping genes both treatment conditions and the untreated state maintained the same direction of regulation as in the disease state (Fig. 8A).

Treatment with ALP modulated the expression of 35 dysregulated genes in *NPHP1*-deficient hURECs, reversing their dysregulation compared to healthy controls. Of these, 24 genes belong to the 'UP-DOWN' set (Fig. 8A), including those associated with immune regulation (*ZFP36*, *SMAD3*, *HLA-DMB*, *CREB5* and *KLF6*), energy metabolism (*ENO2*, *PC*, *PFKFB3* and *VGF*), O-glycosylation (*GALNT9* and *GALNT12*), and the MAPK pathway (*DUSP5* and *DUSP6*), among others. The remaining 11 genes in the 'DOWN-UP' set (Fig. 8B) include those related to the immune response (*C1S*, *PLXDC2* and *CMTM4*), signalling and cholesterol regulation (*PGRMC2*), energy metabolism (*ACSS3*), folate transport (*SLC46A3*), ion transport (*SLC22A17*), prostaglandin metabolism (*MGST1*) and translational fidelity.

Treatment with AG556 modulated the expression of 69 initially dysregulated genes in *NPHP1*-deficient hURECs towards levels comparable to healthy controls. This is evident from the clustering of AG556-treated hURECs with the control samples for both the 'UP-UP' and 'DOWN-DOWN' sets, indicating an almost complete reversal of dysregulation (Fig. 8C).

Among the 69 genes, 43 were from the UP-DOWN set, many of which are involved in immune or inflammatory responses (*EGR2*, *ENO2*, *HLA-DMB*, *KRT15*, *PALM3* and *ZFP36*). Additionally, many of these genes are associated with pathways related to extracellular matrix remodelling and cell adhesion (*CDH4*, *CLDN2*, *CREB5*, *DUSP5*, *FBLN7*, *FER1L4*, *NAB2*, *NAV2*, *NEFH*, *PPFIA4*, *SOX11* and *VGF*). The 26 genes in the DOWN-UP set encompass processes related to these functions as well, including scavenger receptor–ligand binding and folate transmembrane transport (Fig. 8C; Fig. S8B).

When comparing the disease-signature-enhancing gene sets (UP–UP and DOWN–DOWN) across both drug treatments, we observed substantial overlap (Fig. S8C,D). Pathway enrichment analysis of the UP–UP gene set – genes upregulated in *NPHP1*-deficient hURECs and further increased upon both ALP treatment and AG556 – revealed significant enrichment of biological processes related to phospholipid biosynthesis and metabolism (*ATM*, *HMGCS1*, *SMG1*, *PIKFYVE* and *FNIP2*), multicellular organism growth (*ATM*, *PLAG1* and *VPS13A*), chemotaxis (*LYST*, *MOSPD2* and *PIKFYVE*), endosomal and vacuolar transport (*LYST*, *GCC2*, *VPS13A* and *PIKFYVE*), and protein and lipid localization (*GCC2*, *AKAP11*, *VPS13A*, *PIKFYVE*, *STARD4* and *MOSPD2*) (Table S10). Notably, phospholipid metabolism and trafficking are essential for maintaining the lipid composition of the ciliary membrane, which is crucial for signal transduction and ciliary dynamics (Conduit and Vanhaesebroeck, 2020; Stilling et al., 2022; Conduit et al., 2024).

Some genes in the UP-UP sets showed treatment-specific enhancement; among these, cilia-related genes, as reported by SysCilia and CiliaCarta databases, such as *NR1H4*, *AHI1* and *WDR19*, were further upregulated specifically in ALP-treated cells. *NR1H4* encodes the nuclear receptor FXR, a transcription factor known to suppress autophagy and ciliogenesis under nutrient-rich conditions (Lee et al., 2014; Seok et al., 2014; Liu et al., 2018b). *AHI1* encodes Jouberin, a transition zone protein known to be important in the kidney through formation of heterodimers with nephrocystin-1 (Eley et al., 2008) and acts as a bridging molecule between other complexes of NPHP ciliary proteins (Sang et al., 2011). Pathogenic variants in this gene are associated with Joubert syndrome (Valente et al., 2006). *WDR19* encodes a protein in the intraflagellar transport complex, and pathogenic variants are associated with a rare subtype of nephronophthisis called NPHP13 (Tang et al., 2024). The full list of cilia-related genes identified in the DEG analysis is listed in Table S11.

Pathway enrichment analysis of the DOWN-DOWN gene set – genes downregulated in *NPHP1*-deficient hURECs and further decreased by both ALP and AG556 – highlighted stress-adaptive responses and processes essential for cell structure, survival and repair, including actin filament organization, cell junctions, intrinsic apoptosis, endocytosis, membrane projection organization, cell proliferation, tissue morphogenesis and wound healing (Table S11).

## DISCUSSION

Pathogenic variants in *NPHP1* are the most common cause of NPHP (Srivastava and Sayer, 2014). Homozygous whole-gene deletions typically account for ~25% of all *NPHP1* pathogenic variants (Hildebrandt et al., 1997). Aside from the kidney features, which are commonly classified as NPHP and cystic kidney disease, *NPHP1* pathogenic variants cause extrarenal phenotypes, including retinal dystrophy and brain malformations in ~10% of cases (Srivastava et al., 2017a; Esson et al., 2024). Recent research has identified *NPHP1* whole-gene deletions in adult individuals with unexplained kidney failure (Snoek et al., 2018) highlighting that this condition is often underdiagnosed and can lead to adult phenotypes as well as the childhood-onset NPHP phenotypes. The molecular genetic causes of NPHP have implicated defects in the primary cilia as part of the disease pathogenesis. The exact pathophysiology is unresolved; however, initial reports of *NPHP1* pathogenic variants implicated a role in the adherens junction complexes (Hildebrandt et al., 1997) and indeed histological descriptions consistently reveal tubular basement membrane defects.

In this study, we present a unique opportunity to investigate the ciliary phenotype and transcriptomic signature of *NPHP1* through the examination of a family where both parents (I:1, I:2) carry a

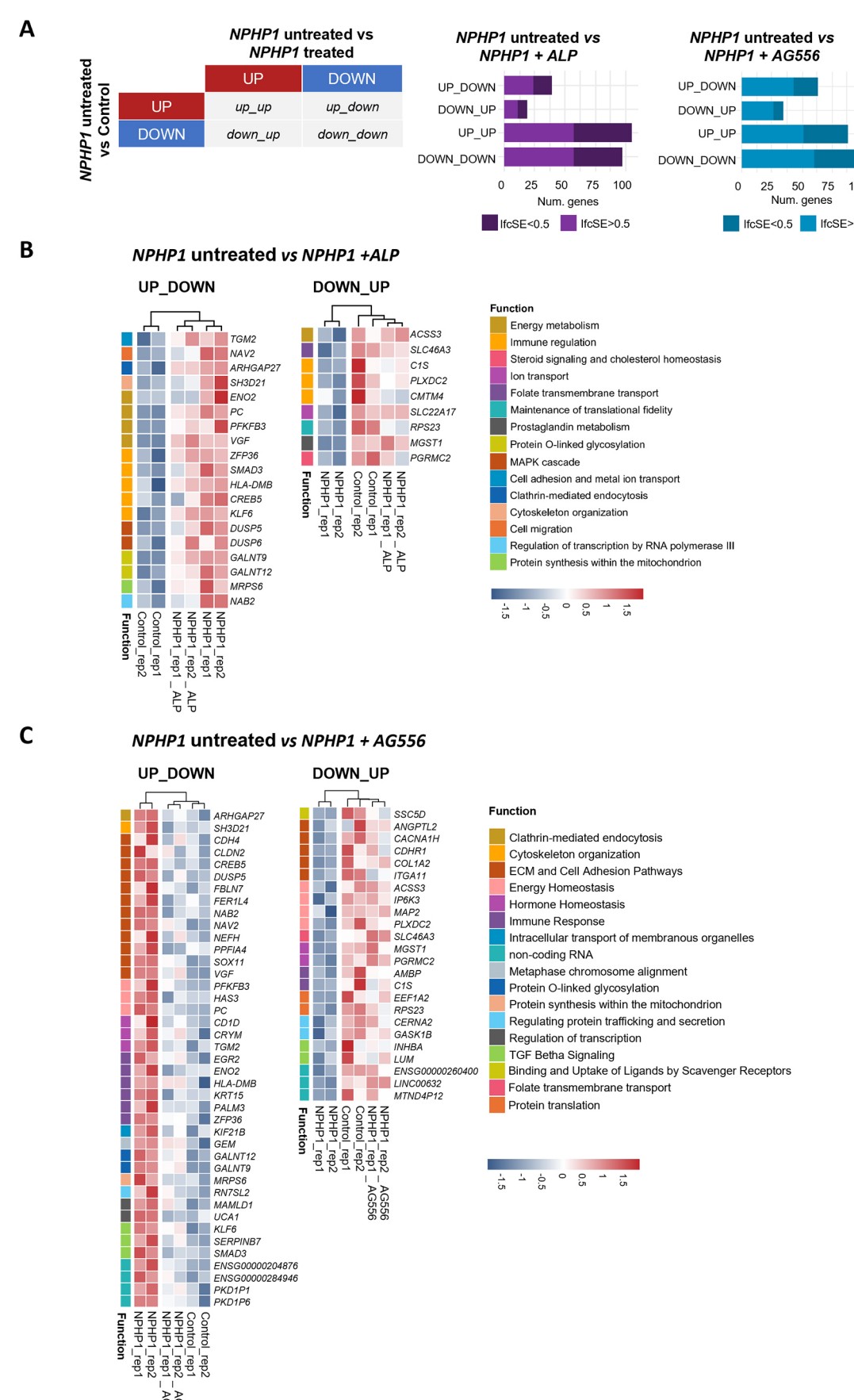

**Fig. 8.** See next page for legend.

**Fig. 8. Treatment effects on transcriptional disease signature.**
(A) Schematic representation of gene sets obtained by overlapping differentially expressed genes from the comparisons of *NPHP1*-untreated versus control, and *NPHP1*-untreated versus *NPHP1*-treated samples (left). Bar plots display the number of genes in each overlapping set between *NPHP1* untreated versus control samples and *NPHP1* untreated versis *NPHP1* plus ALP (purple) or versus *NPHP1* AG556 (blue) samples. (B,C) Heatmap illustrating UP-DOWN and DOWN-UP genes (*P*adj<0.05, lfSE<1, and |log$_2$FC|>0.58) for the comparisons between *NPHP1* untreated versus control samples and (B) *NPHP1* untreated versus *NPHP1* plus ALP or (C) *NPHP1* untreated versus *NPHP1* plus AG556.

heterozygous deletion of the *NPHP1* gene, and their three affected children, two sons (II:1 and II:3) and one daughter (II:2), carry a homozygous deletion of this gene. The diagnosis of this family was possible through the analysis of copy number variants in the whole genome of the proband sequenced via the Genomics England 100,000 genomes project. Although the two older siblings (II;1, II;2) had lost their ability to produce urine due to kidney failure, we had the opportunity of deriving hURECs from urine samples of the younger sibling (II:3) earlier in the disease course, who had not yet reached kidney failure.

Using these hURECs, we phenotyped the renal epithelial cell cilia and confirmed that they recapitulated the ciliary phenotype seen in the biopsy from the kidney biopsy of the affected sibling (II:2). This ciliary phenotype includes an increased length and tortuosity, and a higher percentage of ciliated cells compared to primary hUREC controls. These findings suggest that *NPHP1* whole-gene deletion is associated with consistent alterations in primary cilia morphology across both kidney tissue (including renal collecting duct cells identified by positive staining for aquaporin-2) and the derived renal epithelial cells, characterized by increased cilia length and an enhanced tortuosity.

This is in line with previous observations in both *in vivo* and *in vitro* models of *NPHP1* deficiency. In human induced pluripotent stem cells (hiPSCs) derived from individuals with a *NPHP1* whole-gene deletion, cilia were elongated compared to those seen in hiPSCs from healthy donors. Overexpression of *NPHP1* in these derived cells led to a reduction in cilia length and an increase in the percentage of ciliated cells, indicating a direct role for *NPHP1* in regulating ciliogenesis and cilia morphology (Arai et al., 2024). Similarly, in hiPSCs from healthy donors with a CRISPR-Cas9-mediated deletion of *NPHP1* exon 1, cilia were also longer and more abundant than in unedited controls (Arai et al., 2024). *In vivo*, *NPHP1*-knockout mice show increased cilia length in dilated tubules of the kidney cortex (Garcia et al., 2022), and cells from kidney biopsies from individuals with *NPHP1* variants causing tubular dilatations also displayed both increased cilia length and tortuosity (Garcia et al., 2022). These findings parallel our observations in both II:2 kidney tissue and sibling II:3 hURECs. Interestingly, however, reduced ciliation is seen only in certain contexts, and in a Madin–Darby canine kidney (MDCK) model with shRNA-mediated *Nphp1* knockdown, shortened cilia are noted (Garcia et al., 2022). Notably, this phenotype was partially reversed under 3D culture conditions, with increased cilia length (Delous et al., 2009), underscoring the influence of cellular context and culture conditions on ciliary outcomes. A contrasting case was reported, where a reduction in both cilia number and length in a resected kidney from a child with Senior–Løken syndrome and *NPHP1* deletion was seen, possibly reflecting differences tissue remodelling, or tubular segment affected (Ning et al., 2021). Therefore, care must be taken when interpreting cilia morphology and ciliation defects and in directly comparing primary versus immortalized cell lines and different model systems,

as the baseline ciliary phenotype might vary depending on the experimental system.

Through transcriptomic analysis of primary hURECs from the younger sibling (II:3), we observed upregulation of stress- and injury-related genes (*GALNT9*, *HAVCR1* and *VGF*) and downregulation of genes involved in extracellular matrix organization and fibrosis regulation (*ITGA11*, *THBS1* and *PDGFRB*). Enrichment analysis revealed activation of pathways related to inflammation, actin cytoskeleton dynamics, and nephron development, while pathways involved in basement membrane organization, apoptosis, cell–cell junctions, and focal adhesions were downregulated. More specifically, several RTK-related pathways were altered, with suppression of VEGFA–VEGFR2 and PDGF signalling, and activation of EGF–EGFR, MAPK, and PI3K–AKT pathways. Together, these changes suggest a shift toward pro-proliferative and stress-adaptive states, accompanied by impaired structural maintenance and regulation of fibrosis. Consistent with this, a previous analysis of a biopsy from an individual with a *NPHP1* pathogenic variant showed a significant increase of SOX9 in kidney tubules, suggesting that the cytokine TGF-β, which strongly induces SOX9, is a key mechanism of kidney fibrosis in NPHP (Patel et al., 2025). Indeed, within our *NPHP1*-deletion specific transcriptomic dataset, SOX9 was increased by a fold change of 1.3 compared to control and was reduced by a fold change of 2.29 by EGFR inhibitor AG556 (Tables S1, S6).

Our results align with those previously reported in a mouse model of *Nphp1* deletion (Garcia et al., 2022), where transcriptomic analysis of whole murine kidneys highlighted dysregulation in extracellular matrix organization and cytokine responses, and increased activity in RTK-associated pathways, such as PI3K–AKT, MAPK, and O-linked glycosylation. These findings are consistent with the changes we observed in our *NPHP1*-deficient hURECs.

Inflammation as a signature in *NPHP1* deficiency is further corroborated by previous work (Quatredeniers et al., 2022) that highlighted inflammation as a hallmark of NPHP, with macrophage and neutrophil infiltration observed in NPHP kidneys, along with upregulation of specific cytokines. Our data revealed increased expression of immune-related genes, including *IL6R*, *NFKBIZ* and *NFAT* family members, and *SEMA4A*. These findings suggest heightened immune activation and possible immune cell recruitment driven by altered cytokine signalling.

In terms of proliferative signalling, our results are consistent with previous studies in kidney organoid models. The upregulation of cell cycle-associated genes in epithelial cells from organoids derived from individuals with *NPHP1* deficiency and CRISPR-edited hiPSCs has been reported (Arai et al., 2024), both of which exhibit primary cilia abnormalities and cyst formation. These findings reinforce the connection between *NPHP1* loss and dysregulated cell cycle progression.

The downregulation of pathways related to basement membrane organization, focal adhesions and cell–cell junctions in *NPHP1*-deficient hURECs is in line with the known role of NPHP proteins in maintaining epithelial integrity. NPHP1, together with NPHP4 and INVS, localizes at cell–cell junctions and plays a critical role in their structural integrity and function (Donaldson et al., 2000; Nürnberger et al., 2002; Delous et al., 2009). The disruption of these pathways in our study suggests that the loss of NPHP1 impairs epithelial cohesion and contributes to early cellular dysfunction in NPHP and is consistent with the histological lesions described in kidney tissue.

Given that there are no established treatments for NPHP (including being steroid unresponsive; Wolf and Hildebrandt, 2011) and hURECs from II:3 represent a rare opportunity to study *NPHP1*-deficient kidney cells while the disease is still in a treatable

window, we took this opportunity to assess whether ALP, a drug previously shown to rescue the phenotype in immortalized URECs from individuals with NPHP (Garcia et al., 2022), could also rescue the phenotype of II:3 primary hURECs. However, although the effects of ALP were as previously described by Garcia et al. (elongation of cilia and increased ciliary percentage) (Garcia et al., 2022), these effects were not beneficial in II:3 primary hURECs, which had already elongated cilia and an enhanced percentage of ciliation. Rather, it worsened the ciliary phenotype, highlighting that ALP might not be suitable for all NPHP patients and that immortalized URECs might behave differently. In primary cultures of hURECs from individuals with NPHP, ALP treatment produced an exaggerated ciliary length phenotype, no rescue of cilia tortuosity and an incomplete rescue of the transcriptomic disease signature, urging some caution in the use of this drug as a potential rescue therapy.

As ALP did not rescue the ciliary phenotype, we searched for other targetable pathways identified through our transcriptome analysis. Notably, EGFR emerged as a key regulator of several upregulated pathways in *NPHP1*-deficient hURECs. EGFR is a key regulator of cell proliferation, and previous studies have identified EGFR as a negative regulator of ciliogenesis in proliferating cells (Kasahara et al., 2018). EGFR phosphorylates the ubiquitinase USP8, which stabilizes the trichoplein–Aurora A pathway and suppresses ciliogenesis. Under serum starvation – a standard method to promote ciliogenesis – this EGFR–USP8 signalling is reduced, allowing cilia formation to occur (Kasahara et al., 2018).

Our findings, however, suggest a distinct role for EGFR in serum-starved hURECs derived from II:3. We observed elevated EGFR expression concurrent with robust ciliogenesis. This indicates that in the context of *NPHP1* deficiency, EGFR upregulation might be uncoupled from its canonical anti-ciliogenic function, pointing to a potentially altered regulatory mechanism in these II:3-derived cells. We tested an EGFR inhibitor AG556, which resulted in a reduction of ciliary percentage, length and tortuosity to levels comparable to those of the healthy control, suggesting that abnormal EGFR signalling indeed plays a role in the ciliary phenotype observed in these *NPHP1*-deficient primary cells.

Consistent with the phenotypic rescue, treatment with AG556 modulated the expression of a large number of dysregulated genes in *NPHP1*-deficient hURECs towards levels comparable to healthy controls. This included rescuing immune and inflammatory responses, aberrations in pathways related to extracellular matrix remodelling and cell adhesion. The rescue of disease signature was more evident in response to AG556 treatment than ALP treatment. Both treatments induced extensive gene expression changes, but AG556 exerted a broader corrective effect, shifting a larger number of dysregulated genes towards expression levels observed in healthy controls.

There are some obvious limitations to the data presented. Although primary hURECs cultured in 2D, provide a valuable model for assessing primary ciliary phenotypes in renal epithelial cells, such as cilia length, tortuosity and the percentage of ciliated cells, they do not allow evaluation of more complex renal phenotypes, including tissue organization, basement membrane integrity or cyst formation. Here, we studied hURECs derived from a single NPHP individual, which limits the generalizability of our findings. *NPHP1*-associated kidney phenotypes might vary between individuals or families due to genetic background or modifier genes, and future studies should expand to include samples from multiple affected individuals, perhaps with a range of kidney phenotypes to capture this heterogeneity. Although AG556 treatment rescued certain ciliary phenotypes *in vitro*, it is worth noting that tyrosine kinase inhibitors have been associated with

adverse renal effects *in vivo*. Kidney adverse effects vary by agent and can include acute tubular injury, proteinuria, hypertension and glomerular diseases such as thrombotic microangiopathy (TMA) and focal segmental glomerulosclerosis (FSGS) (Nürnberger et al., 2002; Izzedine et al., 2007), which could limit their therapeutic applicability. Finally, our transcriptomic data show that drug treatments ALP and AG556 both reinforced rather than reversed some of the disease-associated gene expression changes. This suggests that pharmacological monotherapies might be insufficient to restore transcriptional homeostasis and that combinatorial therapeutic strategies or gene therapies might be necessary for more effective rescue. Given the limitations of 2D culture systems in modelling tissue remodelling and cilia function related to mechanical flow and chemical concentration, further *in vivo* or organoid-based studies would be needed to determine whether the identified UP-UP and DOWN-DOWN transcriptional changes reflect adaptive compensation already present in the *NPHP1*-deficient kidney epithelium or true exacerbation of disease processes.

Taken together, our transcriptomic analyses reveal that although both ALP and AG556 induce extensive gene expression changes in *NPHP1*-deficient hURECs, their effects diverge in terms of pathway modulation and potential therapeutic benefit. ALP treatment selectively activates pathways associated with autophagy and lipid metabolism but does not fully restore the disease-associated transcriptional profile. In contrast, AG556 exerts a broader corrective effect, shifting a larger number of dysregulated genes toward expression levels observed in healthy controls, particularly those involved in immune regulation, extracellular matrix remodelling and cell adhesion. However, these results also highlight the persistence of disease-reinforcing expression patterns under both treatments and the complexity of re-establishing transcriptional homeostasis in *NPHP1*-deficient cells.

In conclusion, hURECs from individuals with *NPHP1* deficiency have provided the opportunity to assess ciliary phenotypic and disease transcriptomic signatures of *NPHP1*-deficiency in the kidney. Given an upregulation in EGFR signalling in disease cells and both a phenotypic and transcriptomic rescue following treatment, EGFR inhibition is a potential therapeutic strategy for renal ciliopathies such as NPHP. To rigorously assess its translational relevance, this therapy should be systematically evaluated in established *in vivo* models of NPHP to determine therapeutic efficacy, optimize dosing strategies and evaluate potential adverse effects prior to the initiation of human clinical trials.

## MATERIALS AND METHODS
### Clinical and genomic studies
Genetic, clinical and *in vitro* investigations were carried out on a family with three siblings diagnosed with NPHP. Clinical, imaging and histological data and familial information was obtained by review of clinical records and re-evaluation of pathological specimens using light and immunofluorescence microscopy. The collection of clinical data was undertaken with informed consent and ethical approval was given by the National Research Ethics Service Committee North East – Newcastle and North Tyneside 1 (08/H0906/28+5) and the National Research Ethics Service Committee North East (14/NE/1076). When using human tissues, the authors confirm that all clinical investigations were conducted according to the principles expressed in the Declaration of Helsinki.

The genomic research was conducted with data and findings from the Genomics England 100,000 Genomes Project (Turro et al., 2020). All participants in the 100,000 Genomes Project provided written consent to access their anonymized clinical and genomic data for research purposes. The Genomics England 100,000 Genomes Project was approved by the Health Research Authority Research Ethics Committee East of England –

Journal of Cell Science

Cambridge South (14/EE/1112) (Smedley et al., 2021). Specifically, using a cystic kidney disease virtual gene panel (https://panelapp.genomicsengland.co.uk/panels/283/), whole-genome sequencing (WGS) data from available family members was analysed for disease-causing variants. In addition, available BAM files for affected family members, parents and control individuals were visualized within Integrative Genomics Viewer (IGV) to determine read depth across the *NPHP1* gene region on chromosome 2q13. Genomic research findings of *NPHP1* whole-gene deletion were verified by the Northern Genetics Service diagnostics laboratory using multiplex ligation-dependent probe amplification (MLPA).

## Kidney biopsy methods
Histological analysis of biopsy specimen (for II.2) was processed using standard techniques and stained with periodic acid-Schiff using standard protocols. Tissue was processed and imaged under EM using standard protocols. Immunofluorescence staining for ciliary marker ARL13B was performed using rabbit anti-ARL13B (1:200, Proteintech, 17711–1-AP); and secondary donkey anti-rabbit-IgG Alexa Fluor® 488 (Thermo Fisher Scientific, A-21206) and DAPI (Sigma-Aldrich). Images were captured by a researcher who was not aware of the tissue origin using a Zeiss Axioscope 5 microscope. Length of primary cilia were measured using FIJI (ImageJ) software using the segmented line tool as previously described (Molinari et al., 2020).

## hUREC culture
Human urine-derived renal epithelial cells (hURECs) were isolated from one male NPHP1 individual (II:3) and two healthy, age- and sex-matched male donors as previously described (Srivastava et al., 2017b; Garcia et al., 2022). For each individual, urine samples were collected at two time points 5 months apart, generating two independent hUREC cultures per donor as biological replicates.

For drug-response experiments, each of the two untreated hUREC cultures from individual II:3 was further expanded and treated separately with either Alprostadil (ALP) or AG556, resulting in two biological replicates for each drug treatment.

In total, eight hUREC samples were obtained (see Figs S3 and S6): two untreated cultures from the NPHP1 individual (biological replicates, 5 months apart); two untreated cultures from a healthy control donor (biological replicates, 5 months apart); two cultures from individual II:3 treated with ALP (each derived from one untreated replicate); two cultures from individual II:3 treated with AG556 (each derived from one untreated replicate).

For hUREC isolation, the fresh urine samples were centrifuged at 400 *g* for 10 min to form a urinary sediment pellet. The pellets were washed with phosphate-buffered saline (PBS) containing antibiotics to minimize the risk of contamination and resuspended in primary medium [DMEM F12 (43.6 ml, Gibco), FBS (5 ml; Gibco), 5000 U/ml penicillin/streptomycin (0.7 ml, Thermo Fisher Scientific), 250 μg/ml amphotericin B (0.7 ml, Merck) and 50 μl renal epithelium cell growth medium (REGM) SingleQuots (Lonza, CC-4127)] in 12-well plates. After 72 h medium was replaced by proliferation hUREC medium [renal epithelium basal medium (REBM; Lonza, CC-3191), renal epithelia cell growth medium (REGM) SingleQuots (Lonza, CC-4127) and 2% FBS].

## *In vitro* drug treatment of hURECs
The hURECs were treated with 2 μM ALP (Merk-Millipore, Y0000054) or 0.008% DMSO as vehicle control, and 2 μM AG556 (Tocris, 0616) or 0.02% DMSO for 48 h before analysis. ALP is a synthetic prostaglandin E1 analogue (Leslie et al., 2025) and AG556 is a selective EGFR tyrosine kinase inhibitor (Levitzki and Gazit, 1995).

## hUREC immunofluorescence studies
The hURECs were grown on 24-well plates with cover slips ($2 \times 10^4$ cells per well). After 48 h of serum starvation, the cells were fixed with ice-cold methanol for 10 min and saturated with 5% BSA in TBS for 30 min. The cells were then incubated for 2 h at room temperature with the following primary antibodies: rabbit anti-ARL13B (1:200, Proteintech, 17711-1-AP) and mouse anti-γ-tubulin (1:300, Sigma-Aldrich, T5326). Following washes with 0.1% Tween 20 in TBS, the cells were incubated for 1 h at

room temperature with the following secondary antibodies donkey anti-rabbit-IgG Alexa Fluor 488 (Thermo Fisher Scientific, A-21206); goat anti-mouse-IgG Alexa Fluor 594 (Thermo Fisher Scientific, A-11032). Following final washes in 0.1% Tween 20 in TBS, cells were mounted in Vectashield (Vector Laboratories Ltd, H-1200). Images and *z*-stacks were captured by a researcher who was not aware of the cell origin using a Zeiss Axioscope 5 microscope (Ramsbottom et al., 2018). Measurement of cilia including length and tortuosity measurements were performed using FIJI (ImageJ) software using the segmented line tool to calculate the sum of angles (°) as previously described (Srivastava et al., 2017b; Molinari et al., 2020).

## hURECs bulk RNA-seq
The eight hUREC samples detailed in the 'hUREC culture' section and shown in Figs S4 and S7 were used for bulk RNA sequencing analysis. Cells were cultured to ∼90% confluence and serum starved for 48 h, then dissociated using TrypLE (Gibco, 12604013), and cell pellets were washed once with PBS and stored at −80°C. These samples were sent to Novogene (Novogene Co., UK), a commercial facility, for RNA extraction and sequencing. RNA-seq experiments were performed in biological duplicates.

## Data preprocessing and quality control
Raw sequencing reads were processed by Novogene (Novogene Co., Ltd., UK), including quality trimming, adapter removal, and alignment of clean reads to the human reference genome (GRCh38) using HISAT2 version 2.0.5 (Kim et al., 2019). Quality control of the processed reads was independently verified by us using FastQC version 0.12.1 (https://www.bioinformatics.babraham.ac.uk/projects/fastqc/). The number of clean reads per sample ranged from 35,000,000 to 50,000,000, with a median of 40,171,270 reads.

Data analysis was performed using R version 4.4.0 (https://www.r-project.org/). Gene expression quantification, defined as the number of reads overlapping with exons grouped by gene, was performed using the summarizeOverlaps() function from the GenomicAlignments package version 1.42.0 (Lawrence et al., 2013).

Primary hURECs can contain a mixture of cell types; therefore, to filter genes expressed in the most abundantly represented cell types and ensure that differential expression analysis was conducted in comparable cell types, we utilized an unpublished bulk RNA-seq dataset of 17 hUREC samples generated in our laboratory, including individuals with various kidney diseases as well as healthy controls, as a reference. Using this reference dataset, we applied the following criteria: genes were retained if they had a counts per million (CPM) greater than 1 (equivalent to ∼10 counts for library sizes exceeding 30,000 reads) in at least two samples from this study and in at least five samples from the reference panel.

Data manipulation was performed using dplyr version 1.1.4 (https://dplyr.tidyverse.org), tidyr version 1.3.1 (https://tidyr.tidyverse.org), and reshape2 version 1.9.6 (Wickham, 2007). CPM were calculated using the edgeR package version 4.2.2 (Robinson et al., 2010), and PCA was performed on the log-transformed count data (log2 counts per million+1) using the prcomp() function in R software. The scale.=TRUE parameter was used for data scaling. The top two principal components (PC1 and PC2) were extracted and visualized in a scatter plot using ggplot2 version 3.5.1 (https://ggplot2.tidyverse.org). ComBat function from the (Leek et al., 2012) was employed to correct the raw counts for any batch effects.

## Differential expression and functional enrichment analyses
Differential expression gene (DEG) analyses were carried out using DESeq2 version 1.44.0 (Love et al., 2014). Genes with an adjusted *P*-value of less than 0.05 and an absolute $\log_2$ fold change greater than 1.5 were considered differentially expressed. Upregulated and downregulated genes were identified. A volcano plot was generated using the ggplot2 package version 3.5.1 in R software.

Pathway enrichment analyses were performed using Metascape, version 3.5 (Zhou et al., 2016), which includes Gene Ontology (GO) terms and other relevant databases such as KEGG, Reactome and WikiPathways. Only genes with a *P*-adjusted value of less than 0.05, an absolute $\log_2$ fold change greater than 1.5, and a lfcSE of less than 0.05 were used as input for this

analysis. Network visualization for Biological Process GO term enrichment was generated using the package aPEAR version 1.0.0 (Kerseviciute and Gordevicius, 2023). GO Cellular Component analysis was conducted on the top 100 differentially expressed downregulated genes using the BiNGO plugin in Cytoscape (v3.10.3; Shannon et al., 2003). GO enrichment analysis for Cellular Component terms was performed using the hypergeometric test, with multiple testing correction applied using the Benjamini and Hochberg false discovery rate (FDR) method. Barplot visualizations were generated using ggplot2 package version 3.5.1 in R software. To visualize the expression profiles of the top significant genes with the smallest lfcSE values, heatmaps were generated using the R package pheatmap version 1.0.12 (https://github.com/raivokolde/pheatmap).

### Validation via qPCR

To validate the RNA-seq results (Table S1), a subset of the differentially expressed genes was selected for validation via quantitative PCR (qPCR). RNA was extracted from the hURECs with or without drug treatments using an RNeasy mini kit (Qiagen). 1 µg of RNA was reverse transcribed using SuperScript™ III Reverse Transcriptase (ThermoFisher Scientific, 18080093). The resulting cDNA was diluted 10-fold in nuclease free water to a final concentration of 10 ng/µl. Quantitative analysis of gene expression was carried out using SYBR green qPCR master mix (Thermo Fisher Scientific, 4385618) according to the manufacturer's instructions using gene specific primer pairs (Table S2). The qPCR was performed using QuantStudio™ 7 Flex Real-Time PCR System (Thermo Fisher Scientific, 4485701). Expression levels were normalized to the expression of housekeeping gene *GAPDH*. The qPCR results were compared to the RNA-seq data.

### Web resources

To support variant analysis and interpretation, a combination of bioinformatic tools and databases was employed. The Ensembl Variant Effect Predictor (VEP; https://www.ensembl.org/info/docs/tools/vep/index.html) was used to annotate genetic variants, providing information on their predicted effects on genes, transcripts, and protein sequences. Population-level allele frequency data were obtained from gnomAD v2.1.1 (https://gnomad.broadinstitute.org/), which was used to assess the rarity of the identified variants across diverse populations. To evaluate tissue-specific gene expression relevant to the variants, the Genotype-Tissue Expression (GTEx) database (https://www.gtexportal.org/home/) was utilized. Homozygositymapper (https://www.homozygositymapper.org/) was employed to identify regions of homozygosity within the genome. The Integrative Genomics Viewer (IGV) (https://igv.org/doc/desktop/) was used to visually inspect sequencing reads, confirming variant calls and assessing read depth and quality. For clinical significance and previously reported associations of variants, NCBI ClinVar (https://www.ncbi.nlm.nih.gov/clinvar/) and the Online Mendelian Inheritance in Man (OMIM) database (https://omim.org/) were consulted. Family pedigree analysis was facilitated using the Progeny pedigree tool (https://progenygenetics.com/), which allowed visualization of inheritance patterns and segregation of variants within families. To explore the impact of variants on protein structure and conservation, ProteinPaint (https://viz.stjude.cloud/tools/proteinpaint) and SMART (http://smart.embl-heidelberg.de/) were employed, providing insights into domain-level effects and evolutionary conservation. VarSome® (https://varsome.com/) was used for comprehensive variant interpretation, integrating multiple databases and applying ACMG classification guidelines to support pathogenicity assessments.

### Acknowledgements
We thank the affected individuals, their families, and their physicians who contributed to this study. This research was made possible through access to data in the National Genomic Research Library, which is managed by Genomics England Limited (a wholly owned company of the Department of Health and Social Care). The National Genomic Research Library holds data provided by patients and collected by the NHS as part of their care and data collected as part of their participation in research. The National Genomic Research Library is funded by the National Institute for Health Research and NHS England. The Wellcome Trust, Cancer Research UK and the Medical Research Council have also funded research infrastructure.

### Competing interests
The authors declare no competing or financial interests.

### Author contributions
Conceptualization: P.D.S., E.O., H.M., J.A.S., J.E.A.-G.; Data curation: P.D.S., E.O., K.W., J.E.A.-G.; Formal analysis: P.D.S., E.O., M.T.-H.; Funding acquisition: J.A.S.; Investigation: P.D.S., Z.T.S., H.M., B.D., D.R., J.E.A.-G.; Methodology: H.M., B.D., K.W., D.R., C.C., J.E.A.-G.; Project administration: J.A.S.; Resources: J.A.S.; Supervision: J.A.S.; Validation: J.E.A.-G.; Visualization: E.O., J.E.A.-G.; Writing – original draft: J.A.S., J.E.A.-G.; Writing – review & editing: P.D.S., E.O., Z.T.S., H.M., K.W., D.R., C.C., M.T.-H., J.A.S., J.E.A.-G.

### Funding
P.D.S. is funded by Kidney Research UK (RP_007_20210729). E.O. is supported by Postdoc Mobility-Stipendien of the Swiss National Science Foundation (grants P2ZHP3_195181 and P500PB_206851) and Kidney Research UK (grant Paed_RP_001_20180925). M.T.H. is funded by the Northern Counties Kidney Research Fund (20/01) and the PKD Charity. J.A.S. is funded by LifeArc, Medical Research Council (MR/Y007808/1), Kidney Research UK (Paed_RP_001_20180925, RP_007_20210729), the Northern Counties Kidney Research Fund (20/01) and the European Union's Horizon Europe research and innovation programme and from UKRI under grant agreement No: 101080717 (TheRaCil). Open Access funding provided by Newcastle University. Deposited in PMC for immediate release.

### Data and resource availability
The gene quantification data for all samples, including expression values for each gene across the genome, are publicly available and can be accessed at https://github.com/juearcilaga/hUREC_NPHP1_RNAseq.git

RNA datasets have been submitted to GEO NCBI database, study accession ID GSE301183. The full results of the differential expression analysis are in Table S1 and functional enrichment analysis are in Table S3. All the data supporting the findings of this study are available within the article and its supplementary materials.

### First Person
This article has an associated First Person interview with the first author of the paper.

### Peer review history
The peer review history is available online at https://journals.biologists.com/jcs/lookup/doi/10.1242/jcs.264141.reviewer-comments.pdf

### Special Issue
This article is part of the Special Issue 'Cilia and Flagella: from Basic Biology to Disease', guest edited by Pleasantine Mill and Lotte Pedersen. See related articles at https://journals.biologists.com/jcs/issue/138/20

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
