## [Peer Review File · Journal of Cell Science]

Urinary renal epithelial cells can be used for *NPHP1* phenotyping and a personalized therapeutic strategy

Praveen Dhondurao Sudhindar, Eric Olinger, Zachary T. Sentell, Holly Mabillard, Barbora Dicka, Katrina Wood, Dominic Rutland, Catherine Collins, Marco Trevisan-Herraz, John A. Sayer and Juliana E. Arcila-Galvis
DOI: 10.1242/jcs.264141

Editor: Pleasantine Mill

Review timeline

Original submission:	10 May 2025
Editorial decision:	10 June 2025
First revision received:	29 July 2025
Accepted:	2 August 2025

Original submission

First decision letter

MS ID#: jcs.264141

MS TITLE: Using human urinary-derived renal epithelial cells for deep phenotyping of *NPHP1* deletion and determining personalised response to novel therapeutics

AUTHORS: Praveen Dhondurao Sudhindar; Eric Olinger; Zachary T. Sentell; Holly Mabillard; Barbora Dicka; Katrina Wood; Dominic Rutland; Catherine Collins; Marco Trevisan-Herraz; John A Sayer; Juliana E. Arcila-Galvis

ARTICLE TYPE: Research Article

Dear John, Juliana and team,

Thank you for submitting your article to our Special Issue on Cilia and Flagella for the Journal of Cell Science- we have now reached a decision on the above manuscript.

To see the reviewers' reports and a copy of this decision letter, please go to:

As you will see, the reviewers gave favourable reports but raised some points that will require amendments to your manuscript. I hope that you will be able to carry these out because I would like to be able to accept your paper, depending on further comments from reviewers.

From an editorial stand point, this is a strong and compelling exploratory study into the molecular profiling of homozygous *NPHP1* deletions on patient-derived human urinary epithelial cell (hUREC) cellular and transcriptional phenotypes as a means for screening for small molecule modulators for therapeutic interventions. As noted by both reviewers and this editor, a major limitation is that only one patient is used for deriving cells for this study, as the other older siblings no longer had kidneys capable of generating urine for collection. The limitations are discussed and additional independent validation with genome editing to recreate the variants would be beyond the scope of this project. Some in vivo validation is performed on renal biopsy although whether biopsy regions between affected individual and control are similar similar may be unknown- can the authors look at ciliation rates on collecting tubules for example, to reassure looking at similar structures ei

AQP2+ cells? Presenting the data as rates of ciliation, cilia length and tortuosity for biopsy and hURECs with two sibs and two controls would help strengthen conclusions and claims. In addition to addressing the minor points raised by the two reviewers, editorially I would ask the authors to:

- (1) consistently italicize *in vitro* and *in vivo* throughout the manuscript. Italicize genes for qRT-PCR i.e. GAPDH.
- (2) Separate the unit and the value- 2 μ M Alprostadil
- (3) Not capitalize the species of antibodies- i.e. rabbit anti-ARL13B
- (4) Use the Greek symbol for gamma not γ : γ - tubulin
- (5) Specify with Tween used in washes (20 or 80)
- (6) Add symbol for degree ahead of -80° C
- (7) Clarify in methods what is biological duplicates if only one individual is used for bulk sequencing. Similarly explicitly define what are 'replicates' in the results section 'Transcriptomic analysis.... between replicates'. Events quantified are given in the legends but not how many samples these came from- please clarify.
- (8) Data availability- we would strongly encourage you to please upload RNA sequencing data and read me files to GEO (<https://www.ncbi.nlm.nih.gov/geo/info/submission.html>) for all samples used in this paper. Add the accession numbers to the data availability statement in this paper.
- (9) Explain tortuosity and how it is quantified.
- (10) The term neddylation should not be capitalized.
- (11) Clearly define what the deletion breakpoints are for maternal and paternal deletion alleles- are they the same? Could be included under pedigree in Fig 1A.
- (12) Add in ciliation rates in Fig 2- possibly looking only at collecting tubules to help give segment identity. Fig 6 move percentage ciliation ahead of length and tortuosity- be consistent in data presentation and text description.

Reviewer 1

Suthindar et al. present a comprehensive molecular and ciliary characterization of renal epithelial cells (hURECs) derived from patient urine of individuals with a homozygous NPHP1 deletion. The study investigates the molecular alterations that occur in hURECs and identifies relevant signaling pathways and biological processes associated with nephronophthisis (NPH). The study's approach is well-structured, and the authors provide convincing evidence to support the utility of hURECs as a model for understanding the molecular mechanisms of NPH.

Key Findings:

1. **Comparable Ciliary Alterations in hURECs and Kidney Tissue:**
The authors demonstrate that molecular ciliary changes in hURECs are similar to those observed in directly biopsied kidney tissue. The direct comparison between samples from two siblings strongly supports the potential of hURECs as a relevant model for studying the molecular underpinnings of nephronophthisis.
2. **Identification of Upregulated and Downregulated Signaling Pathways:**
Multiple signaling pathways and biological processes are identified as being either upregulated or downregulated in the context of a homozygous NPHP1 deletion. This provides valuable insights into the molecular alterations associated with NPH.
3. **Potential Role of EGFR Signaling Pathway in NPH Pathogenesis:**
The study identifies the EGFR signaling pathway as potentially relevant to the pathogenesis of NPH, offering new avenues for therapeutic exploration.
4. **Reversibility of Molecular and Cellular Alterations Through Pharmacological Intervention:**
The study demonstrates the ability to reverse or correct observed molecular and cellular changes through pharmacological intervention in the cell model. This is an important step toward potential therapeutic approaches for NPH.

The study is well-organized, and the findings are presented in a clear, logical progression. The methodology is rigorous, and the data support previous findings and studies, both in animal models and in human URECs.

Limitations:

1. **Sample Size and Generalizability:**

A key limitation of the study is that the results are based on hURECs from a single individual and compared to samples from only one control. This raises concerns about the generalizability of the findings. It would be beneficial to expand the sample size and include more control samples to strengthen the conclusions and ensure the findings are not individual-specific. The authors appropriately address this limitation in the discussion.

2. Characterization of control individual:

Is the healthy control a heterozygous parent or a sex- and age-matched individual?

3. Lack of Detailed Explanation for Observed Dysregulation:

The study reports that the dysregulation of certain genes and signaling pathways is not corrected by the administration of either ALP or AG556, but is instead exacerbated (up-up and down-down). While the authors speculate on potential reasons for these observations, further investigation and explanation would be beneficial. Additional data or potential mechanistic explanations could enhance the depth of the findings.

Questions to the authors:

1) Regarding the clinical presentation of the affected siblings:

* At least two of them carry the diagnosis autism. Do the authors think that this is explained or a part of the NPHP1 phenotypic spectrum?

* Proband II:1 presented at the age of 17 years with ESKD (GFR 5 ml/min/1,73m²).

Surprisingly his Hb was still 10.6 g/dl at the time? Do the authors have an explanation for this observation as early anemia is a typical feature of NPH?

* Why did the genetic analysis of proband II:2 not detect the homozygous NPHP1 deletion in the first place?

2) Within the genes of the Down-Up set „steroid signalling“ is mentioned. Do the authors know of any successful treatment intervention with steroids in either an in-vitro or in vivo setting of NPH?

3) Regarding the different behaviour of UREC cilia under ALP treatment: Do the authors have an explanation, why cilia in the French study (Garcia, Serafin 2022; URECS from multiple patients) responded differently to ALP treatment than the observations made in URECs of the individual proband II:2?

4) Figure 2 and 3: Cilia length in hURECs seems significantly longer than cilia length in kidney biopsies for both, NPHP1 patients and controls. The same applies for tortuosity. Is there an explanation for this?

5) Figure 8: In the setting of ALP as well as AG556 treatment, the majority of regulated genes either belong to the up-up or down-down set, while only in a smaller amount of genes responds in an up-down or down-up way. How does this observation correspond with the potential use of these components as therapeutic agents?

Minor issues:

- page 4, line 29 till the end of the paragraph „We characterized the response of ALP treatment...to human studies“. To avoid redundancy I would suggest to transfer this paragraph into the discussion section and delete it from the introduction.

- page 18 line 9-20: I would also suggest to leave this section to discussion.

- page 8, line 55 „CPM“ is mentioned as an abbreviation here already, while the explanation "counts per million" only follows on the following page.

- discussion, page 19, line 4: „Homozygous whole gene deletions typically account for 25% of all NPHP1 pathogenic variants“. This is not correct. Depending on the actual cohort, homozygous NPHP1 deletions either account for 25% of all NPHP variants. Regarding all NPHP1 variants, the homozygous deletion is found in the vast majority of cases.

Typos:

-page 4, line 25: hURECs instead of HURECs.

-page 13, line 34: the word „during“ makes no sense from e at this place.

-page 20, line 24: „... cilia were shorter in standard culture were perviously noted.“ This sentence makes no sense.

-page 20, line 59: „that“ instead of „thar“

Overall Assessment:

The study by Suthindar et al. provides compelling evidence for the utility of hURECs as a model for investigating the molecular mechanisms of nephronophthisis. The findings are well-supported by the data and are of considerable interest to the field. The study has notable strengths in its methodology and potential therapeutic implications but is limited by the small sample size and lack of detailed mechanistic insights into certain findings. With these limitations in mind, I recommend that the manuscript be accepted with minor revisions.

Reviewer 2**SUMMARY OF THE ADVANCE MADE IN THIS PAPER AND ITS POTENTIAL SIGNIFICANCE TO THE FIELD**

Sudhindar et al. report the identification of a mutation in NPHS1 in a patient with nephronophthisis and their family members. They present findings from the investigation of ciliary phenotypes in patient-derived human ureteric epithelial cells (hURECs) and RNA sequencing (RNA-seq) analysis comparing patient and control transcripts. Pathway analysis pointed to alterations in EGFR signaling.

To explore therapeutic interventions, the authors used a drug identified in a 2D drug screen to rescue ciliogenesis in NPHS1-depleted cells. They used alprostadil and an EGFR kinase inhibitor (AG556)—the latter selected based on their transcriptomic data—as candidate drugs. These compounds were tested on hURECs to assess their effects on ciliary structure and transcriptome profiles. Treatment with AG556 resulted in significantly better rescue of ciliary phenotypes compared to alprostadil. However, transcriptomic analysis indicated only minimal restoration of gene expression profiles following treatment, with widespread dysregulation persisting.

The authors emphasize that while patient-derived cells serve as a valuable model for studying disease mechanisms and potential therapies, *in vivo* studies are still needed to validate these findings. This study highlights a novel approach to drug discovery by leveraging transcriptomic analysis of patient hURECs to inform therapeutic strategies for NPHS1-associated nephronophthisis.

SUGGESTIONS TO AUTHORS

1. Fig S3B and Fig S6A b why is there such a variance between samples with the same genetic background?
2. Fig 6H authors state that "ALP treatment led to and increase in ... tortuosity" but this is not what the graph shows. Graph shows that there is a significant difference between control and NPHS+DMSO, but no difference between NPHS+DMSO and NPHS+ALP. Furthermore, datapoints look the same as in figure 3H which compares untreated control and patients cells, yet here the same datapoints are used for DMSO treated patient cells (for both graphs in figure 3G and H). And then some the same datapoints are being used in figure 7 H and I (control) as in figure 6 H and I. Personally, I think authors should put figures 3,6 and 7 together for a good overview of untreated vs treated cells rather than drop feed it.
3. "Treatment of NPHP1-deficient hURECs from patient II:3 with AG556 led to a reduction in the percentage of ciliated cells, cilia length and tortuosity, restoring these parameters to levels comparable to those observed in control hURECs (Figure 7)." that is not correct for percentage ciliation.
4. Figure 7J was not discussed by authors at all. It is a great figure, visually representing the important results of author's work. Please discuss.
5. "When comparing the disease signature with drug-induced changes, we found that most overlapping genes in the ALP condition were upregulated in the disease and became even more upregulated upon treatment (UP-UP group, Figure 8A). Conversely, in the AG556 condition, overlapping genes were generally downregulated in disease and became further downregulated with treatment (DOWN-DOWN group, Figure 8A)." based on the graphs I would say it's a very close call. Would authors want to support that statement with numbers?
6. "While serum starvation induces a stress-adaptive state, the further suppression of these pathways suggests limited compensatory capacity, with treatments potentially amplifying

dysfunction rather than reversing it." do authors mean here that treatment with either ALP or AG556 is not only not beneficial but also harmful? Where transcriptomic analysis done in serum starved cells?

7. Authors show cilia phenotype reverse after patient cells treatment, what about cilia signalling, which is more functional representation of cell state? "These findings suggest that NPHP1 whole gene deletion is associated with consistent alterations in primary cilia morphology across both kidney tissue and patient-derived renal epithelial cells, characterized by increased cilia length and an enhanced tortuosity." and increased percentage of ciliated cells? Admittedly, this result is missing in figure 1, but it is kind of addressed in Figure 6I (though patients cells were treated with DMSO, which in itself can have effect on ciliation)?

8. Although authors specify number of events that are then taken for statistical calculations it is unclear how many biological replicates were investigated?

Editorial changes:

P5 line HURECs to hUREC; *in vivo* to *in vivo*

Figure S6B is discussed before S6A, please rearrange to keep order.

Between call for Fig8A and Fig8C is Fig8B which is mislabelled as Fig8A.

Please change "Joubert Syndrome" to Joubert syndrome

First revision

Author response to reviewers' comments

Thank you for the helpful reviews and editorial comments. We have taken these suggestions and provided a robust response to all of the points raised. We submit a revised manuscript and a point by point response is given below.

Editors comments

From an editorial stand point, this is a strong and compelling exploratory study into the molecular profiling of homozygous *NPHP1* deletions on patient-derived human urinary epithelial cell (hUREC) cellular and transcriptional phenotypes as a means for screening for small molecule modulators for therapeutic interventions. As noted by both reviewers and this editor, a major limitation is that only one patient is used for deriving cells for this study, as the other older siblings no longer had kidneys capable of generating urine for collection. The limitations are discussed and additional independent validation with genome editing to recreate the variants would be beyond the scope of this project. Some *in vivo* validation is performed on renal biopsy although whether biopsy regions between affected individual and control are similar may be unknown- can the authors look at ciliation rates on collecting tubules for example, to reassure looking at similar structures in AQP2+ cells?

Yes we have performed an additional analysis of ciliation rates and cilia morphology in collecting duct segments identified by AQP2+ IF staining and present the data in a new supplemental figure, Figure S2. This shows increases in length and tortuosity in this nephron segment. We have added this new quantification to Figure S2 and have expanded the results and discussion to include this new data. The measurements are comparable to the renal biopsy data shown in Figure 2.

Presenting the data as rates of ciliation, cilia length and tortuosity for biopsy and hURECs with two sibs and two controls would help strengthen conclusions and claims.

Yes we agree, but only one patient underwent a renal biopsy and as stated 2 siblings have now received a renal transplant, therefore hURECs do not represent the patient's kidney disease.

In addition to addressing the minor points raised by the two reviewers, editorially I would ask the authors to:

(1) consistently italicize *in vitro* and *in vivo* throughout the manuscript. Italicize genes for qRT-PCR i.e. *GAPDH*

Corrected

(2) Separate the unit and the value- 2 μ M Alprostadil

Corrected

(3) Not capitalize the species of antibodies- i.e. rabbit anti-ARL13B

Corrected

(4) Use the Greek symbol for gamma not γ : γ - tubulin

Corrected

(5) Specify with Tween used in washes (20 or 80)

Corrected

(6) Add symbol for degree ahead of -80 °C

Corrected

(7) Clarify in methods what is biological duplicates if only one individual is used for bulk sequencing. Similarly explicitly define what are 'replicates' in the results section 'Transcriptomic analysis.... between replicates'. Events quantified are given in the legends but not how many samples these came from- please clarify.

We have revised the results paragraph mentioned as well as the methods sections, "Human Urine Derived Renal Epithelial Cell (hUREC) Culture" and "Bulk RNA Sequencing of hURECs," to clarify the definitions of biological duplicates and replicates. Bulk RNA sequencing was performed using hURECs derived from one NPHP1 patient and one healthy, age- and sex-matched control individual. For each individual, two independent hUREC cultures were established from urine samples collected at two separate time points five months apart. These time-separated cultures serve as biological replicates.

(8) Data availability- we would strongly encourage you to please upload RNA sequencing data and read me files to GEO (<https://www.ncbi.nlm.nih.gov/geo/info/submission.html>) for all samples used in this paper. Add the accession numbers to the data availability statement in this paper.

We have now uploaded the datasets to the GEO NCBI database, the study accession ID is GSE301183. The data availability statement now reads:

The gene quantification data for all samples, including expression values for each gene across the genome, are publicly available and can be accessed at https://github.com/juearcilaga/hUREC_NPHP1_RNAseq.git
RNA datasets have been submitted to GEO NCBI database, study accession ID GSE301183.

(9) Explain tortuosity and how it is quantified.

We measure tortuosity by manually drawing the region of interest (ROIs) in ImageJ using segmented line tool to trace along the cilia. Then we use a macro within Image J to get sum of angles for each cilium, which is the proxy for tortuosity and is what is plotted in the final graph. We have added details to the methods.

(10) The term neddylation should not be capitalized.

Corrected

(11) Clearly define what the deletion breakpoints are for maternal and paternal deletion alleles- are they the same? Could be included under pedigree in Fig 1A.

We have now clarified the current resolution of the deletion breakpoints. All genomic coordinates refer to the hg38 reference genome. In the proband and both parents, the deletion begins downstream of exon 2 of the MALL gene, removing exon 1, and includes the entire gene bodies of NPHP1 and MTLN. Based on our data, the deletion extends at least from approximately chr2:110,090,979 to chr2:110,229,113.

The regions flanking this deletion appear as low-mappability regions in the UCSC hg38 Genome Browser. Upstream, low mappability extends toward chr2:109,931,849, and downstream, it extends to around chr2:110,441,473. These regions overlap with duplications (low-copy repeats). The presence of these segmental duplications prevents precise determination of the exact breakpoints in either parent. Therefore, we cannot definitively conclude whether the maternal and paternal deletion alleles are identical in size or position.

We have included this clarification under the legend for Figure 1E in the revised manuscript and added a reference to Figure S1 illustrating the segmental duplications.

(12) Add in ciliation rates in Fig 2- possibly looking only at collecting tubules to help give segment identity. Fig 6 move percentage ciliation ahead of length and tortuosity- be consistent in data presentation and text description.

There was no change in ciliation rates between control and patient on the renal biopsy analysis in Figure 2 and new figure S2 which identifies collecting duct (aquaporin-2 positive cells. We have added this information to the text. We have corrected Figure 6 to be consistent.

Reviewer 1: Sudhinder et al. present a comprehensive molecular and ciliary characterization of renal epithelial cells (hURECs) derived from patient urine of individuals with a homozygous NPHP1 deletion. The study investigates the molecular alterations that occur in hURECs and identifies relevant signaling pathways and biological processes associated with nephronophthisis (NPH). The study's approach is well-structured, and the authors provide convincing evidence to support the utility of hURECs as a model for understanding the molecular mechanisms of NPH.

Key Findings:

1. Comparable Ciliary Alterations in hURECs and Kidney Tissue:

The authors demonstrate that molecular ciliary changes in hURECs are similar to those observed in directly biopsied kidney tissue. The direct comparison between samples from two siblings strongly supports the potential of hURECs as a relevant model for studying the molecular underpinnings of nephronophthisis.

2. Identification of Upregulated and Downregulated Signaling Pathways:

Multiple signaling pathways and biological processes are identified as being either upregulated or downregulated in the context of a homozygous NPHP1 deletion. This provides valuable insights into the molecular alterations associated with NPH.

3. Potential Role of EGFR Signaling Pathway in NPH Pathogenesis:

The study identifies the EGFR signaling pathway as potentially relevant to the pathogenesis of NPH, offering new avenues for therapeutic exploration.

4. Reversibility of Molecular and Cellular Alterations Through Pharmacological Intervention:

The study demonstrates the ability to reverse or correct observed molecular and cellular changes through pharmacological intervention in the cell model. This is an important step toward potential therapeutic approaches for NPH.

The study is well-organized, and the findings are presented in a clear, logical progression. The methodology is rigorous, and the data support previous findings and studies, both in animal models and in human URECs.

Thank you for the positive review and summary of main findings.

Limitations:

1. Sample Size and Generalizability:

A key limitation of the study is that the results are based on hURECs from a single individual and compared to samples from only one control. This raises concerns about the generalizability of the findings. It would be beneficial to expand the sample size and include more control samples to strengthen the conclusions and ensure the findings are not individual-specific. The authors appropriately address this limitation in the discussion.

Yes, we agree that this would be very desirable. For this family the other 2 affected individuals received a renal transplant and we were unable to obtain hURECs that were representative of their native kidneys. In future studies we agree to use a range of NPHP1 patients to look for disease signature across different patients.

2. Characterization of control individual:

Is the healthy control a heterozygous parent or a sex- and age-matched individual?

The healthy control is a sex and age matched individual unrelated to the family. We have clarified this in the text.

3. Lack of Detailed Explanation for Observed Dysregulation:

The study reports that the dysregulation of certain genes and signaling pathways is not corrected by the administration of either ALP or AG556, but is instead exacerbated (up-up and down-down). While the authors speculate on potential reasons for these observations, further investigation and explanation would be beneficial. Additional data or potential mechanistic explanations could enhance the depth of the findings.

Yes, thank you for the important comment. We agree that the transcriptomic data prompts additional mechanistic work. We have validated some of the signalling pathway changes in selected pathways using qPCR and that is included within the manuscript (Figure 4B). We plan in future studies to assess ciliary protein content in disease and in response to therapies but this is beyond the scope of the present study which aims to act as a proof of concept for the use of hUREC transcriptomics as a disease signature tool.

Questions to the authors:

1) Regarding the clinical presentation of the affected siblings:

* At least two of them carry the diagnosis autism. Do the authors think that this is explained or a part of the NPHP1 phenotypic spectrum?

Yes, we agree, a **whole gene deletion of NPHP1** has been associated with both **nephronophthisis type 1 (NPHP1)** and **neurodevelopmental disorders**, including **autism spectrum disorder (ASD)**. We have added a comment and a reference. (Sakakibara N, et al Comprehensive genetic analysis using next-generation sequencing for the diagnosis of nephronophthisis-related ciliopathies in the Japanese population. *J Hum Genet.* 2022 Jul;67(7):427-440.)

* Proband II:1 presented at the age of 17 years with ESKD (GFR 5 ml/min/1,73m²). Surprisingly his Hb was still 10.6 g/dl at the time? Do the authors have an explanation for this observation as early anemia is a typical feature of NPH?

We agree, in **NPHP**, **early-onset anaemia** is indeed a well-documented and expected finding, often preceding the decline in GFR. The hemoglobin (Hb) level of **106 g/l in kidney failure** is considered relatively **preserved**, and **unusual**. The decline in renal function was probably **gradual**, and compensatory mechanisms (e.g. increased erythropoietin) might have maintained Hb higher. Also, dehydration may have cause a haemoconcentrating effect at presentation.

* Why did the genetic analysis of proband II:2 not detect the homozygous NPHP1 deletion in the first place?

This is a good point. A negative report was issued by Genomic England following analysis of WGS data. The CNV analysis pipelines were inadequate during the early phases of the Genomic England Rare Disease project and these have since been improved. The initial analysis was performed on vcf files, and because the gene was absent no variant were noted.

2) Within the genes of the Down-Up set „steroid signalling" is mentioned. Do the authors know of any successful treatment intervention with steroids in either an in-vitro or in vivo setting of NPH? Whether this upregulation of steroid signaling with treatment could be therapeutically relevant. I did not find evidence of this but I am not confident enough to say no one has done it, do you know of anyone doing this? And specifically, whether steroids have ever been used successfully (in vitro or in vivo) in NPH models.

This is a good point. As noted, in our manuscript, we highlight inflammation as part of the disease signature in NPHP1-deficient hURECs, and that steroid signaling is enriched in the Down-Up gene set (i.e., downregulated in disease, upregulated upon treatment). However, there is no evidence in vitro or in vivo for the use of corticosteroids for the treatment of NPHP. We have added a comment in the revised manuscript to clarify this.

3) Regarding the different behaviour of UREC cilia under ALP treatment: Do the authors have an explanation, why cilia in the French study (Garcia, Serafin 2022; URECS from multiple patients) responded differently to ALP treatment than the observations made in URECs of the individual proband II:2?

We thank the reviewer for this thoughtful question. As described in the manuscript (Results, page 19; Discussion, page 26), the study by Garcia et al. (2022) reported that ALP treatment elongated cilia and increased the percentage of ciliated cells in immortalised URECs from NPHP patients. In our study, we observed the same treatment effect in primary hURECs from NPHP1 patients – ALP increased both cilia length and ciliation rate.

The key difference lies in the baseline ciliary phenotype. Garcia et al. reported shortened cilia in their immortalised URECs at baseline, whereas our primary hURECs showed elongated cilia prior to treatment. The reason for this discrepancy remains unclear, but we propose two possible explanations:

Different genetic backgrounds – our primary hURECs and the immortalised lines in Garcia et al. may represent distinct patient populations.

Impact of cell immortalization – Garcia et al. do not report the ciliary phenotype prior to immortalization, so it is unknown whether the observed shortening is disease-related or a result of the immortalization process itself.

As mentioned in the Discussion (page 24), even within the same study by Garcia et al., ciliary phenotypes differ across model systems. For example, in NPHP1 knockout mice and kidney biopsies from NPHP1 patients with tubular dilatations, they observed elongated and tortuous cilia, which is consistent with our findings in primary hURECs. This suggests that the phenotype may vary depending on the model. Also on page 24, we mention that our findings align with Arai et al. (2024), who reported elongated cilia in hiPSCs derived from NPHP1-deleted patients. Similar elongation has been observed in 3D MDCK cyst models (Delous et al., Hellman et al.). However, contrasting findings have also been reported. For example, Ning, Song et al. (2021) described reduced cilia number and length in a resected kidney from a child with Senior-Løken syndrome and NPHP1 deletion.

In summary, while ALP consistently increases cilia length and ciliation across models, the baseline ciliary phenotype may vary depending on the experimental system. We acknowledge that the precise causes of this variability remain unknown and have highlighted this in the Discussion to avoid overgeneralization. We have made additional comments within the manuscript to reflect and clarify these arguments.

4) Figure 2 and 3: Cilia length in hURECs seems significantly longer than cilia length in kidney biopsies for both, NPHP1 patients and controls. The same applies for tortuosity. Is there an explanation for this?

Kidney biopsy tissues from patients provide a complex microenvironment for cell to cell and cell to extracellular matrix interactions while the cultured cells typically lack these interactions, which can affect the cilia length and tortuosity. However, we were able to recapitulate the directional changes in cilia length and tortuosity in hURECs compared the native kidney biopsies from patient and control. The huREC cells are serum starved in vitro which could also make the cilia longer when compared to native kidney renal epithelial cell cilia.

5) Figure 8: In the setting of ALP as well as AG556 treatment, the majority of regulated genes either belong to the up-up or down-down set, while only in a smaller amount of genes responds in a up-down or down-up way. How does this observation correspond with the potential use of these components as therapeutic agents?

We appreciate the reviewer's insightful comment. As noted in the manuscript (page 21), most gene expression changes in response to ALP and AG556 fall into the disease-signature-enhancing categories (UP-UP and DOWN-DOWN), rather than the reversing categories (UP-DOWN and DOWN-UP). This suggests that the treatments may not broadly reverse the disease signature but rather reinforce certain trends.

After reviewing the text, we noted that our use of the term "disease-enhancing" could be misleading, as it suggests only the amplification of disease-related dysfunction, rather than the enhancement of disease-related dysfunction. However, this reinforcement could also reflect adaptive or protective responses in NPHP1-deficient cells. We have updated this to "disease-signature-enhancing" for clarity.

Given the limitations of 2D culture systems in modelling tissue remodelling and cilia function related to mechanical flow and chemical concentration, further in vivo or organoid-based studies would be needed to determine whether these changes reflect adaptive compensation already present in the NPHP1 patient or true exacerbation of disease processes. We have now expanded our discussion of this within the revised document.

Minor issues:

- page 4, line 29 till the end of the paragraph „We characterized the response of ALP treatment....to human studies“. To avoid redundancy I would suggest to transfer this paragraph into the discussion section and delete it from the introduction.

Corrected

- page 18 line 9-20: I would also suggest to leave this section to discussion.

Thank you, we have moved this section to the discussion.

- page 8, line 55 „CPM“ is mentioned as an abbreviation here already, while the explanation "counts per million" only follows on the following page.

Corrected

- discussion, page 19, line 4: „Homozygous whole gene deletions typically account for 25% of all NPHP1 pathogenic variants“. This is not correct. Depending on the actual cohort, homozygous NPHP1 deletions either account for 25% of all NPHP variants. Regarding all NPHP1 variants, the homozygous deletion is found in the vast majority of cases.

Thank you for this comment, we have corrected the sentence to ensure accuracy.

Typos:

-page 4, line 25: hURECs instead of HURECs.

Corrected

-page 13, line 34: the word „during“ makes no sense for me at this place.

Corrected

-page 20, line 24: „...cilia were shorter in standard culture were perviously noted.“ This sentence makes no sense.

Corrected

-page 20, line 59: „that“ instead of „thar“

Corrected

Overall Assessment:

The study by Suthindar et al. provides compelling evidence for the utility of hURECs as a model for investigating the molecular mechanisms of nephronophthisis. The findings are well-supported by the data and are of considerable interest to the field. The study has notable strengths in its methodology and potential therapeutic implications but is limited by the small sample size and lack of detailed mechanistic insights into certain findings. With these limitations in mind, I recommend that the manuscript be accepted with minor revisions.

Thank you for the helpful review.

Reviewer 2:

Sudhindar et al. report the identification of a mutation in NPHS1 in a patient with nephronophthisis and their family members. They present findings from the investigation of ciliary phenotypes in patient-derived human ureteric epithelial cells (hURECs) and RNA sequencing (RNA-seq) analysis comparing patient and control transcripts. Pathway analysis pointed to alterations in EGFR signaling.

To explore therapeutic interventions, the authors used a drug identified in a 2D drug screen to rescue ciliogenesis in NPHS1-depleted cells. They used alprostadil and an EGFR kinase inhibitor (AG556)—the latter selected based on their transcriptomic data—as candidate drugs. These compounds were tested on hURECs to assess their effects on ciliary structure and transcriptome profiles. Treatment with AG556 resulted in significantly better rescue of ciliary phenotypes compared to alprostadil. However, transcriptomic analysis indicated only minimal restoration of gene expression profiles following treatment, with widespread dysregulation persisting.

The authors emphasize that while patient-derived cells serve as a valuable model for studying disease mechanisms and potential therapies, *in vivo* studies are still needed to validate these findings. This study highlights a novel approach to drug discovery by leveraging transcriptomic analysis of patient hURECs to inform therapeutic strategies for NPHS1-associated nephronophthisis.

SUGGESTIONS TO AUTHORS

1. Fig S3B and Fig S6A b why is there such a variance between samples with the same genetic background?

Although these are urine derived samples from one patient with same genetic background there is variance as we show in the PCA plots Fig S3B (now Fig S4B) and Fig S6A (now Fig S7AB). This variance can mostl likely be attributed to biological factors. Fresh urine samples were collected from the patient and hURECs were isolated at two different time points. We made sure the samples were taken at the same time of the day and underwent RNA seq alongside control samples in an identical way. Variation is likely to have arisen due to the cellular heterogeneity i.e variations in proportion of different cell subtypes in hURECs collected at different time points and the progression of the kidney disease in the patient. Importantly, although there was inter-sample variability, the replicates from subject and control were separated, underscoring the robustness of the disease-specific transcriptional signature we generated.

2. Fig 6H authors state that "ALP treatment led to and increase in ... tortuosity" but this is not what the graph shows. Graph shows that there is a significant difference between control and NPHS+DMSO, but no difference between NPHS+DMSO and NPHS+ALP.

Furthermore, datapoints look the same as in figure 3H which compares untreated control and patients cells, yet here the same datapoints are used for DMSO treated patient cells (for both graphs in figure 3G and H). And then some the same datapoints are being used in figure 7 H and I (control) as in figure 6 H and I. Personally, I think authors should put figures 3,6 and 7 together for a good overview of untreated vs treated cells rather than drop feed it.

Thank you, we have taken this comment very seriously. We have reanalysied data contributing to Fig 3,6, 7 and ensured that appropriate control data sets have been used to compare with patient results. Revised figure panels are now included in Figure 3 G,H,I and Figure 7 GH,I. We have still separated the figures, rather than combine all the date, so that the data can be clearly shown in a narrative way before and after treatments as used in these model systems.

You are correct, Fig 6H - now 6I shows a non-significant difference in tortuosity between NPHP+DMSO and NPHP1+ALP, we have corrected the text to make this clearer.

3. "Treatment of NPHP1-deficient hURECs from patient II:3 with AG556 led to a reduction in the percentage of ciliated cells, cilia length and tortuosity, restoring these parameters to levels comparable to those observed in control hURECs (Figure 7)." that is not correct for percentage ciliation.

We agree that the statement is not true for percentage ciliation as there is a significant difference between the control hURECs and patient II:3 hURECs treated with AG556. We have corrected this.

4. Figure 7J was not discussed by authors at all. It is a great figure, visually representing the important results of author's work. Please discuss.

Thank you we have now added into the text a discussion of Figure J.

5. "When comparing the disease signature with drug-induced changes, we found that most overlapping genes in the ALP condition were upregulated in the disease and became even more upregulated upon treatment (UP-UP group, Figure 8A). Conversely, in the AG556 condition, overlapping genes were generally downregulated in disease and became further downregulated with treatment (DOWN-DOWN group, Figure 8A)." based on the graphs I would say it's a very close call. Would authors want to support that statement with numbers?

Thank you for the comment, we have rephrased the interpretation of the Figure 8A and the numbers are given in the paragraphs following this and table S10. For ALP treatment 24 genes belong to the UP-DOWN group and 11 genes in the DOWN-UP group. For the AG556 treatment 43 are in the UP-DOWN group and 26 are in the DOWN-UP group.

6. "While serum starvation induces a stress-adaptive state, the further suppression of these pathways suggests limited compensatory capacity, with treatments potentially amplifying dysfunction rather than reversing it." do authors mean here that treatment with either ALP or AG556 is not only not beneficial but also harmful? Where transcriptomic analysis done in serum starved cells?

Thank you for the comment, To avoid confusion or overinterpretation, we have removed the sentence in question from the manuscript. hURECs were serum starved to induce ciliogenesis.

7. Authors show cilia phenotype reverse after patient cells treatment, what about cilia signalling, which is more functional representation of cell state?

"These findings suggest that NPHP1 whole gene deletion is associated with consistent alterations in primary cilia morphology across both kidney tissue and patient-derived renal epithelial cells, characterized by increased cilia length and an enhanced tortuosity." and increased percentage of ciliated cells? Admittedly, this result is missing in figure £, but it is kind of addressed in Figure 6l (though patients cells were treated with DMSO, which in itself can have effect on ciliation)?

Thank you. We have clearly shown a rescue of cilia morphology. A rescue of ciliary signalling has not been directly shown in our studies, rather we focused on the ability for a change in transcriptome to be detected following treatment. We plan functional studies in future studies. DMSO 0.02% was used as a vehicle control for drug studies. DMSO at low concentrations is generally considered safe and is widely used as a solvent in cilia studies. Higher concentrations of DMSO >0.5-1% can disrupt ciliogenesis, shorten cilia and affect cilia protein trafficking. We did not observe these effect with 0.02% DMSO.

8. Although authors specify number of events that are then taken for statistical calculations it is unclear how many biological replicates were investigated?

Thank you. As outlined above for reviewer 1, we have revised the results paragraph mentioned as well as the methods sections, "Human Urine Derived Renal Epithelial Cell (hUREC) Culture" and "Bulk RNA Sequencing of hURECs," to clarify the definitions of biological duplicates and replicates. Bulk RNA sequencing was performed using hURECs derived from one NPHP1 patient and one healthy, age- and sex-matched control individual. For each individual, two independent hUREC cultures were established from urine samples collected at two separate time points five months apart. These time-separated cultures serve as biological replicates.

Editorial changes:

P5 line HURECs to hUREC; in vivo to in vivo

Corrected

Figure S6B is discussed before S6A, please rearrange to keep order.

Corrected

Between call for Fig8A and Fig8C is Fig8B which is mislabelled as Fig8A.

Corrected

Please change "Joubert Syndrome" to Joubert syndrome

Corrected

Second decision letter

MS ID#: jcs.264141R1

MS Title: Using human urinary-derived renal epithelial cells for deep phenotyping of NPHP1 deletion and determining personalised response to novel therapeutics

Authors: Praveen Dhondurao Sudhindar; Eric Olinger; Zachary T. Sentell; Holly Mabillard; Barbora Dicka; Katrina Wood; Dominic Rutland; Catherine Collins; Marco Trevisan-Herraz; John A Sayer; Juliana E. Arcila-Galvis

Article Type: Research Article

Dear John, Praveen and team,

I am very pleased to inform you that your revised manuscript has been accepted for publication in Journal of Cell Science, pending standard publication integrity checks. It was accepted on 02 Aug 2025. Where referee reports on this version are available, they are appended below.

Note that when scanning the corrections I noticed that there is a minor typo in the new Figures 6 G-I and 7 G-I where the 1 in NPHP1 has been dropped in the x-axis labels. Currently reads 'NPHP + DMSO' versus 'NPHP1 + drug'.